# Carrier lifetime enhancement in halide perovskite via remote epitaxy

Jie Jiang [1,2,9], Xin Sun [3,9], Xinchun Chen [4], Baiwei Wang [2], Zhizhong Chen[2], Yang Hu[2], Yuwei Guo[2], Lifu Zhang[2], Yuan Ma[5], Lei Gao [5], Fengshan Zheng [6], Lei Jin[6], Min Chen [7], Zhiwei Ma[7], Yuanyuan Zhou[7], Nitin P. Padture [7], Kory Beach [3], Humberto Terrones [3], Yunfeng Shi[2], Daniel Gall [2], Toh-Ming Lu[3], Esther Wertz [3], Jing Feng[1] & Jian Shi [2,8]

Crystallographic dislocation has been well-known to be one of the major causes responsible for the unfavorable carrier dynamics in conventional semiconductor devices. Halide perovskite has exhibited promising applications in optoelectronic devices. However, how dislocation impacts its carrier dynamics in the 'defects-tolerant' halide perovskite is largely unknown. Here, via a remote epitaxy approach using polar substrates coated with graphene, we synthesize epitaxial halide perovskite with controlled dislocation density. First-principle calculations and molecular-dynamics simulations reveal weak film-substrate interaction and low density dislocation mechanism in remote epitaxy, respectively. High-resolution transmission electron microscopy, high-resolution atomic force microscopy and Cs-corrected scanning transmission electron microscopy unveil the lattice/atomic and dislocation structure of the remote epitaxial film. The controlling of dislocation density enables the unveiling of the dislocation-carrier dynamic relation in halide perovskite. The study provides an avenue to develop free-standing halide perovskite film with low dislocation density and improved carried dynamics.

[1] Department of Materials Science and Engineering, Kunming University of Science and Technology, Kunming, Yunnan 650093, China. [2] Department of Materials Science and Engineering, Rensselaer Polytechnic Institute, Troy 12180, United States. [3] Department of Physics, Applied Physics and Astronomy, Rensselaer Polytechnic Institute, Troy, NY 12180, USA. [4] State Key Laboratory of Tribology, Tsinghua University, Beijing 100084, China. [5] Beijing Advanced Innovation Center for Materials Genome Engineering, Institute for Advanced Materials and Technology, University of Science and Technology Beijing, Beijing 100083, China. [6] Ernst Ruska-Centre for Microscopy and Spectroscopy with Electrons and Peter Grünberg Institute, Forschungszentrum Jülich, Jülich, Germany. [7] School of Engineering, Brown University, Providence, RI 02912, USA. [8] Center for Materials, Devices, and Integrated Systems, Rensselaer Polytechnic Institute, Troy, NY 12180, United States. [9] These authors contributed equally: Jie Jiang, Xin Sun. Correspondence and requests for materials should be addressed to L.G. (email: gaolei@ustb.edu.cn) or to J.F. (email: vdmzsfj@qq.com) or to J.S. (email: shij4@rpi.edu)

Dislocation, a crystallographic defect in a crystal structure, is well-known to strongly affect the material's physical properties. Electron-hole recombination mediated via dislocations (e.g., threading dislocation) is one of the predominant loss mechanisms for the sub-optimum performance in semiconductors devices. Typical examples include the quantum efficiency loss in GaN-based light-emitting diodes (LEDs)[1,2] and GaAs solar cells[3]. The relationship between dislocation and carrier dynamic has been well experimentally investigated and understood in conventional semiconductors such as GaN[4] and Si[5]. Recently, halide perovskite has been showing great promises[6,7] in many semiconductor devices such as photovoltaics[8–10], LEDs[11,12], lasers[13–15], photodetectors[16,17] and transistors[18,19]. Despite their 'defects-tolerant' nature[20–22], the effects of grain boundary[23–27], interfaces[16,28,29], points defects[8,30–32], and phase impurity[33–35] in halide perovskite on the carrier dynamics and device performances have been widely recognized and studied[31,36–39]. However, its dislocation-carrier dynamic relation has never been unveiled. Whether dislocation could severely disturb halide perovskite's carrier dynamics has been remaining a puzzle. The lack of dislocation-carrier dynamic study has been mainly due to the inability in controlling dislocation configurations and quantifies.

Recently, monolayer graphene was introduced as a buffer layer for epitaxial growth of semiconductor materials such as GaAs[40,41]. With this concept, remote heteroepitaxy was achieved in the systems of AlN film on sapphire[42], copper film on sapphire[43], and ZnO film on GaN[44]. In these studies, it is believed that the graphene buffer layer screens the majority of the substrate electrostatic potential but still allows a weak substrate-film coupling[45,46]. In addition to graphene transparency, the strong polarity of substrates that provides long-distance electrostatic decaying potential is believed to be the other substantial factor enabling remote epitaxy[40,41]. In this report, by taking advantages of the concept of remote epitaxy and its ability in controlling film-substrate coupling and further regulating dislocation densities (misfit and then threading), we have successfully epitaxially grown halide perovskite crystals and films of controlled dislocations (derived from the density of etching rosettes) on two strong polar NaCl and CaF$_2$ substrates with monolayer graphene buffered. Density functional theory (DFT) calculations have revealed the structure and magnitude of the incompletely screened electrostatic potential from the polar substrates, supporting the remote epitaxy in the present case. The regulated film-substrate interactions have further reflected themselves in controlling the wavelengths of the ferroelastic domains. Molecular-dynamics (MD) simulations reveal the kinetic process during remote epitaxy. Comparing to the ionic epitaxy with high dislocation density (both misfit and threading), the film grown via remote shows much enhanced photoluminescence intensity and increased carrier lifetime. Our successful demonstration of remote epitaxy in halide perovskite provides an approach to develop free-standing halide perovskite film with reduced dislocation density. More importantly, dislocations and their impacts on carrier dynamics and device performance in halide perovskite have to be recognized and scrutinized.

## Results

**Epitaxy relation and lattice structure.** Before growth, graphene layers are transferred onto NaCl(001) and CaF$_2$(001) as substrates (Gr/NaCl(001) and Gr/NaCl(001)) for remote epitaxy. Optical images in Supplementary Fig. 1a and e show the surface morphology of Gr/NaCl(001) and Gr/NaCl(001). Both defected region[47] (low intensity ratio of 2D and G bands in Supplementary Fig. 1c and g, and high intensity of D and G bands in

Supplementary Fig. 1d and h) and good region (high intensity ratio of 2D and G bands in Supplementary Fig. 1c and g, and low intensity ratio of D and G bands in Supplementary Fig. 1d and h) of graphene on substrates are characterized by Raman spectra and mappings, as shown in Supplementary Fig. 1b–d and f–h. The atomic-scale structures of graphene surface on both NaCl(001) and CaF$_2$(001) substrates are investigated by high resolution atomic force microscopy (HRAFM), as shown in Supplementary Fig. 2a and c, respectively. Supplementary Fig. 2b and d show the fast Fourier transforms (FFT) and the bright spots can be indexed to graphene structure. The spots splitting in Supplementary Fig. 2b and the unclear additional spots in Supplementary Fig. 2d could be induced by the chemical potential influence from the substrates underneath.

Chemical vapor deposition (CVD) method was used to achieve both ionic and remote epitaxy (more details can be found in Methods, Characterizations in Supplementary Information). Under optimized growth condition, two typical samples were grown, including CsPbBr$_3$ thin film grown on NaCl(001) substrate (CsPbBr$_3$/NaCl, i.e., ionic epitaxy) and CsPbBr$_3$ flakes grown on monolayer graphene buffered NaCl(001) substrate (CsPbBr$_3$/Gr/NaCl, i.e., remote epitaxy). The crystallinity and the epitaxial relationship of CsPbBr$_3$ with substrates were characterized by X-ray diffraction (XRD) high-resolution reciprocal space mapping (RSM). Figure 1a, b shows the RSM of both 224 peaks for as-grown CsPbBr$_3$ and NaCl substrate in two typical samples, CsPbBr$_3$/NaCl and CsPbBr$_3$/Gr/NaCl. Two types of peaks of CsPbBr$_3$ for both samples could be found (Fig. 1a, b and Supplementary Fig. 3: XRD $\omega$-2$\theta$ scans for both samples). Based on the four-fold symmetry exhibited in the X-ray pole figure in Fig. 1c, the crystal structure for the as-grown samples could be assigned to tetragonal structure (P4/mbm) or orthorhombic structure (Pbnm or Pmna) due to the close lattice constants of $a$ (8.207 Å) and $b$ (8.255 Å) in orthorhombic (Pbnm) phase[48,49]. It should be noted that, for convenience, in some analysis, CsPbBr$_3$ is indexed with pseudocubic structure. The epitaxial relationships for both samples are consistent and could be interpreted as out-of-plane CsPbBr$_3$(001)∥NaCl(001) and in-plane CsPbBr$_3$[100]∥NaCl[100]. The lattice constants of CsPbBr$_3$ are calculated to be $a_\parallel = 5.811$ Å, $a_\perp = 5.830$ Å and $a_\parallel = 5.817$ Å, $a_\perp = 5.850$ Å for CsPbBr$_3$/NaCl and CsPbBr$_3$/Gr/NaCl, respectively. As shown in the insets of Supplementary Fig. 3, high crystallinity for both samples in terms of rocking curves of 004 peaks is obtained even on the multi-domains NaCl substrates. The better crystallinity and the decrease in the full width at half maximum (FWHM) of the rocking curves in the remote epitaxial sample comparing to the ionic epitaxial one (from 0.39° to 0.29°) were characterized. The in-plane orientations of the remote and ionic epitaxial samples can also be revealed by X-ray pole figures of CsPbBr$_3$ 111 and NaCl 111, as shown in Fig. 1c and Supplementary Fig. 4, respectively.

Although the same growth conditions have been applied for both samples, the remote epitaxial one exhibits totally different morphology (flakes), as can be seen in the scanning electron microscope (SEM) image Fig. 1e, due to the remarkable change of the growth thermodynamics and kinetics for remote epitaxy. The flake size varies from less than 1 μm to more than 10 μm, due to the successive epitaxial nucleation which is typical for the halide perovskite epitaxial growth[50]. The size of CsPbBr$_3$ flake and thickness of thin film can be tuned by growth time and substrate temperature. Typical cross-sectional SEM images of the ionic epitaxial film with thickness of around 1 μm and the remote epitaxial films with thickness of around 1.5 μm, 1 μm and 2.2 μm, are shown in Supplementary Fig. 5a, Fig. 1f, Supplementary Fig. 5b, c, respectively. Atomic force microscopy (AFM) images of Supplementary Fig. 6a, b show smooth surface of ionic and remote epitaxial flakes with surface root mean square (RMS)

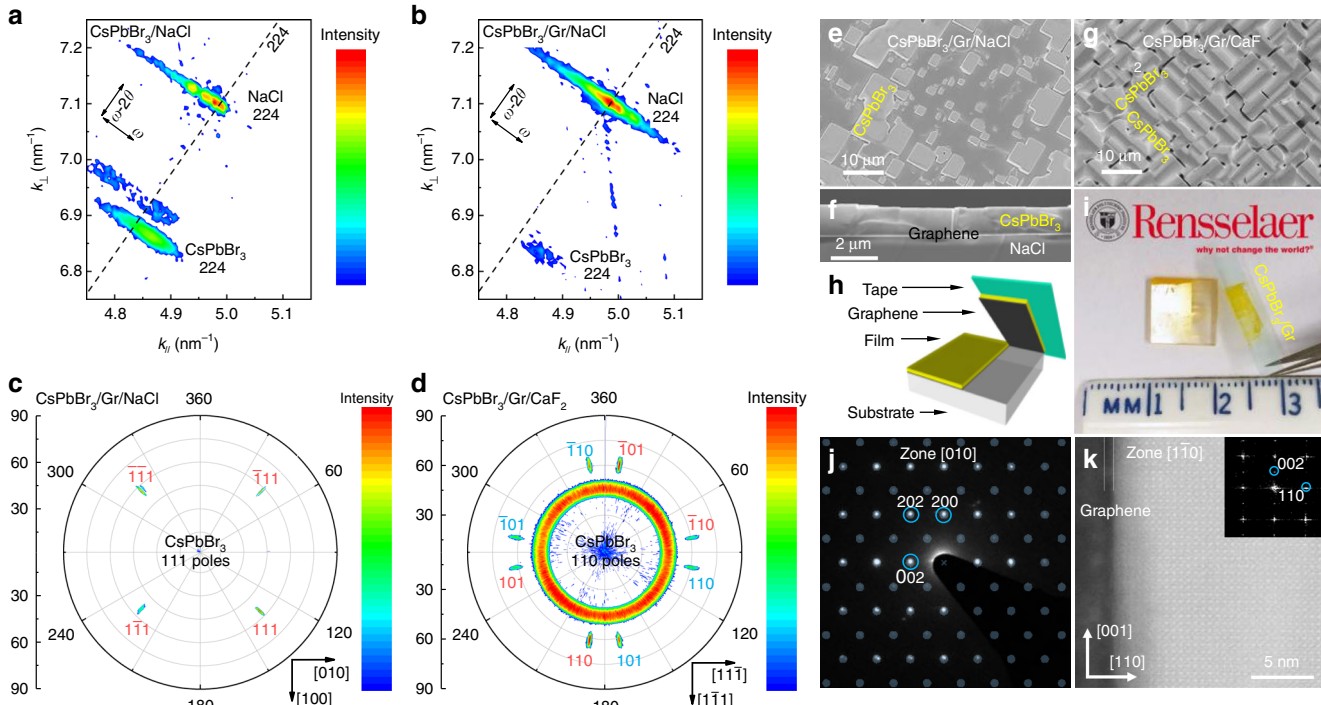

**Fig. 1** Structure and morphology analysis of remote epitaxial CsPbBr$_3$. **a** RSM of 224 peaks for CsPbBr$_3$ thin film and NaCl from ionic epitaxy (CsPbBr$_3$/NaCl). **b** RSM of 224 peaks for CsPbBr$_3$ flakes and NaCl from remote epitaxy (CsPbBr$_3$/Gr/NaCl). **c** X-ray pole figure of CsPbBr$_3$ 111 from CsPbBr$_3$/Gr/NaCl. **d** X-ray pole figure of CsPbBr$_3$ 110 from CsPbBr$_3$/Gr/CaF$_2$. **e** SEM image of remote epitaxial CsPbBr$_3$ flakes on Gr/NaCl. **f** Cross-sectional SEM image of the remote epitaxial CsPbBr$_3$ thin film on Gr/NaCl. **g** SEM image of remote epitaxial CsPbBr$_3$ triangular prisms on Gr/CaF$_2$. **h** Schematic illustration of the exfoliation process on the remote epitaxial sample. **i** Photograph of the as-grown remote epitaxial thin film and exfoliated thin film. **j** Diffraction pattern of remote epitaxial CsPbBr$_3$ (white dots) with simulated pattern on it (transparent bluish dots). **k** Cs-corrected STEM image of remote epitaxial CsPbBr$_3$ and FFT in the inset. (**a–d** are indexed by pseudocubic structure to simplify the analysis, **k** and **j** are indexed by CsPbBr$_3$ orthorhombic structure)

roughness of 0.9 nm and 0.4 nm at 3 μm lateral scale, respectively. The thickness of ionic and remote epitaxial flakes are estimated to be around 750 nm and 860 nm from the height profiles of Supplementary Fig. 6e, f, respectively. The surface morphology and height profiles of both ionic and remote epitaxial films are shown in Supplementary Fig. 6c, g and d, h, respectively. The surface RMS roughness for the ionic epitaxial film in Supplementary Fig. 6c is calculated to be around 4.6 nm at 10 μm lateral scale, while it decreases to 1.5 nm for the remote epitaxial film in Supplementary Fig. 6d.

The as-grown CsPbBr$_3$ flakes in remote epitaxy and the status of graphene after growth are characterized by Raman spectrum and mapping, as shown in Supplementary Fig. 7b and c–f, respectively. The Raman mapping region ($10 \times 10$ μm$^2$) is indicated in green square in Supplementary Fig. 7a. The Raman peaks at around 68 cm$^{-1}$, 120 cm$^{-1}$, and 307 cm$^{-1}$ confirm the orthorhombic phase of CsPbBr$_3$[51], as shown in Supplementary Fig. 7b, c. After growth, the graphene G-band and 2D-band can still be seen in Raman spectra in Supplementary Fig. 7b. Besides, additional D-band of graphene can be found under CsPbBr$_3$ flakes, which might be related to the disorder induced by epitaxial growth. Supplementary Fig. 7f shows a slightly shift towards high wavenumber in 2D-band at the region under flakes, indicating compression in graphene due to epitaxial strain.

Besides the NaCl substrate, the remote epitaxy of CsPbBr$_3$ has also been achieved on CaF$_2$(001) substrate (CsPbBr$_3$/Gr/CaF$_2$), as indicated from the X-ray pole figure of CsPbBr$_3$ 110 in Fig. 1d and φ-scan of CsPbBr$_3$ 110 and CaF$_2$ 111 in Supplementary Fig. 8. SEM image of Fig. 1g and optical image of Supplementary Fig. 9a show the morphology of the epitaxial triangular prisms, revealing two 90°-rotated domains epitaxially grown on

Gr/CaF$_2$(001). The epitaxial relationship between two kinds of epitaxial domains and the substrate is characterized by X-ray pole figure of CsPbBr$_3$ 110 and φ-scan of CsPbBr$_3$ 110 and CaF$_2$ 111, which is out-of-plane CsPbBr$_3$(011)||CaF$_2$(001) and in-plane CsPbBr$_3$[100]||CaF$_2$[010] or CsPbBr$_3$[0$\bar{1}$1]||CaF$_2$[010].

The as-grown epilayer can be easily exfoliated. The exfoliation process is shown in the schematic illustration in Fig. 1h and the corresponding photograph is shown in Fig. 1i. The exfoliated sample can be transferred on to transmission electron microscopy (TEM) grid. TEM of halide perovskite materials is a challenge due to degradation under electron beam[52]. Before degradation, TEM diffraction pattern is obtained, as shown in Fig. 1j. The diffraction spots can be well indexed to support the orthorhombic structure, and the zone axis is calculated to be [010]. The cross-sectional interfacial morphologies and crystallite structures were evaluated by low-voltage Cs-corrected scanning transmission electron microscope (STEM). The STEM samples of CsPbBr$_3$/Gr/NaCl were prepared by focused ion beam (FIB) and details are shown in Methods. Figure 1k demonstrates smooth interfaces between CsPbBr$_3$ (bright part), NaCl (dark part) and Gr (dark line in between) and atomic-resolution image of epilayer CsPbBr$_3$ for CsPbBr$_3$/Gr/NaCl. The FFT in the inset of Fig. 1k confirms the orthorhombic phase of CsPbBr$_3$. More atomic-scale images of epilayer CsPbBr$_3$ and their FFTs are shown in Supplementary Fig. 10a, b and their insets, respectively. It is noted that NaCl in Fig. 1k is amorphous because NaCl is extremely unstable even under low-voltage electron beam, as shown in Supplementary Fig. 11a–c with holes formed. Comprehensive study on stability of both CsPbBr$_3$ and NaCl was carried out by in-situ TEM, as shown in Supplementary Fig. 12 for CsPbBr$_3$/Gr/NaCl and Supplementary Fig. 13 for NaCl in another region. The variation of spots in

FFTs in the insets of Supplementary Figs. 12 and 13 indicate both $CsPbBr_3$ and NaCl are unstable and $CsPbBr_3$ is slightly better than NaCl. Supplementary Movies 1, 2 and 3 show the detail amorphization processes from bright-field images of $CsPbBr_3$/Gr/NaCl and NaCl and diffraction pattern of NaCl, respectively.

**Unscreened polar substrate electrostatic potential**. To understand the physical mechanism for the experimental observation of remote epitaxy in halide perovskite, DFT calculations were performed firstly. For $CsPbBr_3(001)$ growth on Gr/NaCl(001), relaxed four layers of $CsPbBr_3(001)$ lattice on NaCl(001) and monolayer graphene-coated NaCl(001) were chosen, as shown in Supplementary Fig. 14a, b. The interlayer distance between $CsPbBr_3(001)$ and NaCl(001) is 3.0 Å. The interlayer distances between $CsPbBr_3(001)$ and graphene, and between graphene and NaCl(001) are both 3.1 Å. Figure 2a, b present the corresponding charge transfer distributions between $CsPbBr_3(001)$ and NaCl (001), and between $CsPbBr_3(001)$ and NaCl(001) with monolayer graphene insertion. Comparing Fig. 2a, b, we can observe that the insertion of graphene does not change the tendency of charge redistribution between $CsPbBr_3(001)$ and NaCl(001). The characteristics of the interfacial interactions between $CsPbBr_3(001)$ and NaCl(001), and between $CsPbBr_3(001)$ and monolayer graphene buffered NaCl(001) are reflected by their respective interfacial interaction energies in Fig. 2c. The atomic stacking between top layer NaCl(001) and graphene is presented in Fig. 2d. The electrostatic potential distribution contributed by both NaCl and graphene on the surface is shown in Fig. 2e. The observed pattern of blue spots is consistent with NaCl(001) atomic stacking in Fig. 2d, implying the influence of NaCl(001) on the orientation of growth $CsPbBr_3$.

For $CsPbBr_3(011)$ growth on Gr/$CaF_2(001)$, the relaxed three layers of $CsPbBr_3(011)$ lattice on monolayer graphene-coated $CaF_2(001)$ was chosen, as shown in Supplementary Fig. 14c. The interlayer distances between $CsPbBr_3(011)$ and graphene, and between graphene and $CaF_2(001)$ are 2.75 Å and 2.26 Å, respectively. Figure 2f presents the corresponding charge transfer distribution between $CsPbBr_3(011)$ and $CaF_2(001)$ with monolayer graphene intercalation. From Fig. 2f we can observe that the $CaF_2(001)$ substrate influences the charge distribution of graphene and subsequently modulates the growth of $CsPbBr_3(011)$. Then the interfacial interaction energy between $CsPbBr_3(011)$ and monolayer graphene-coated $CaF_2(001)$ was calculated as $-27.83$ meV Å$^{-2}$, indicating the graphene intercalation could effectively reduce the interfacial strain. The atomic stacking between top layer $CaF_2(001)$ and graphene is presented in Fig. 2g. The electrostatic potential distribution contributed by the Ca atoms on the surface directly above graphene, with a distance of 2.7 Å is shown in Fig. 2h. In Fig. 2h, the observed pattern of blue spots is consistent with $CaF_2(001)$ atomic stacking in Fig. 2g, implying the influence of $CaF_2(001)$ on the orientation of growth $CsPbBr_3(011)$.

**Suppressing nucleation and promoting growth in remote epitaxy**. Based on the calculated interfacial free energies in both ionic and remote epitaxy, a semi-quantitative epitaxial nucleation and growth model is proposed. The schematic illustration in Fig. 3a describes the atomistic nucleation process at the initial stage of an ionic epitaxy and a remote epitaxy. Having the monolayer graphene on the substrate (right side of Fig. 3a), the adsorption energy, $E_{ad}$, and the diffusion energy, $E_d$, are completely reduced, thereby affecting the adsorption and diffusion of atoms on the substrate surface. The weakened surface electrostatic potential after graphene screening and reduced adatom-substrate interaction in the case of remote epitaxy would lead to large critical

nuclei, while the stronger ionic bond in ionic epitaxy leads to smaller nuclei, as sketched in purple adatoms in Fig. 3a. In addition, the extremely low diffusion barrier for adatoms on graphene in remote epitaxy makes the adatoms much easier to diffuse to the existing nuclei. Therefore, large size of nuclei with limited quantity is expected for remote epitaxy, as illustrated in Fig. 3a.

The nucleation rate could be calculated based on the classical nucleation theory. Assuming monolayer 2D disc-shaped nuclei at the initial state of growth, the Gibbs free energy of formation for nuclei is given by:

$$\Delta G_i = -\pi r^2 a_0 \Delta\mu + \pi r^2 (\gamma_{int} + \gamma_c - \gamma_s) + 2\pi r a_0 \gamma_c, \quad (1)$$

where $a_0$ is taken to be the lattice constant of $CsPbBr_3$ with respect to the ideal perovskite, $\Delta\mu$ is the difference in volumetric Gibbs free energy of the two phases (solid and gas) (unit: meV Å$^{-3}$) and a general description of supersaturation for $\Delta\mu$ is used, $\gamma_{int}$ is the interfacial free energy (29.0 meV Å$^{-2}$ and 21.8 meV Å$^{-2}$ from DFT calculation for ionic and remote epitaxy as shown in Fig. 2c, respectively), $\gamma_c$ is the surface free energy of $CsPbBr_3$ (76.3 meV Å$^{-2}$, approximated from the bond energy of 248 KJ mol$^{-1}$ for Pb-Br and 389 KJ mol$^{-1}$ for Cs-Br[53], $\gamma_s$ is the surface free energy of substrate (90.3 meV Å$^{-2}$ for ionic epitaxy approximated from the bond energy of 412 KJ mol$^{-1}$ for Na-Cl[53] and 2.9 meV Å$^{-2}$ for remote epitaxy based on the surface free energy of 48 mJ m$^{-2}$ for graphene). The critical nucleus size $r^*$ can be obtained by maximizing $\Delta G_i$ with respect to $r$, which is given by:

$$r^* = \frac{\gamma_c}{\Delta\mu - (\gamma_{int} + \gamma_c - \gamma_s)/a_0}. \quad (2)$$

Thus, the critical Gibbs free energy of formation for a monolayer 2D disc-shaped nucleus is given by:

$$\Delta G_i^* = \frac{\pi a \gamma_c^2}{\Delta\mu - (\gamma_{int} + \gamma_c - \gamma_s)/a_0} = \pi r^* a_0 \gamma_c. \quad (3)$$

Hence, the nucleation rate can be expressed as:

$$N = \Omega\exp\left[-\frac{\Delta G_i^*}{k_B T}\right] = \Omega\exp\left[-\frac{\pi a \gamma_c^2}{k_B T[\Delta\mu - (\gamma_{int} + \gamma_c - \gamma_s)/a_0]}\right] \quad (4)$$

where $\Omega$ is the pre-exponential factor, which is not very much dependent upon the supersaturation and a typical value of $10^{17}$ cm$^{-2}$ s$^{-1}$ is taken for condensation from vapor in our case, $k_B$ is the Boltzmann constant, and $T$ is the temperature (500 K was taken). The nucleation rates in both ionic and remote epitaxy are plotted as a function of supersaturation, as shown in Fig. 3b. The remote epitaxy shows orders of magnitude decrease in nucleation rate comparing to the ionic epitaxy and the decrease even reaches four orders of magnitude at low supersaturation.

Guided from our previous study on ionic epitaxy of halide perovskite[54], the supersaturation for the epitaxial growth could be well controlled by the substrate temperature. In the present work, a temperature gradient on a single substrate has been introduced, leading a supersaturation gradient and different morphologies of halide perovskites on the substrate. Based on the nucleation rates predicted in Fig. 3b, a schematic illustration of a growth model of $CsPbBr_3$ epitaxially grown on NaCl substrate with graphene covered half area is presented in Fig. 3c. For ionic epitaxy, high supersaturation leads the growth of epitaxial thin film, while low supersaturation leads the growth of epitaxial flakes. For remote epitaxy, at the same supersaturation, the nuclei are much sparser than that in epitaxial thin film for the ionic epitaxy, resulting in the growth of epitaxial flakes. The SEM images in Fig. 3d–h and the optical microscope images in Supplementary Fig. 9b, c show

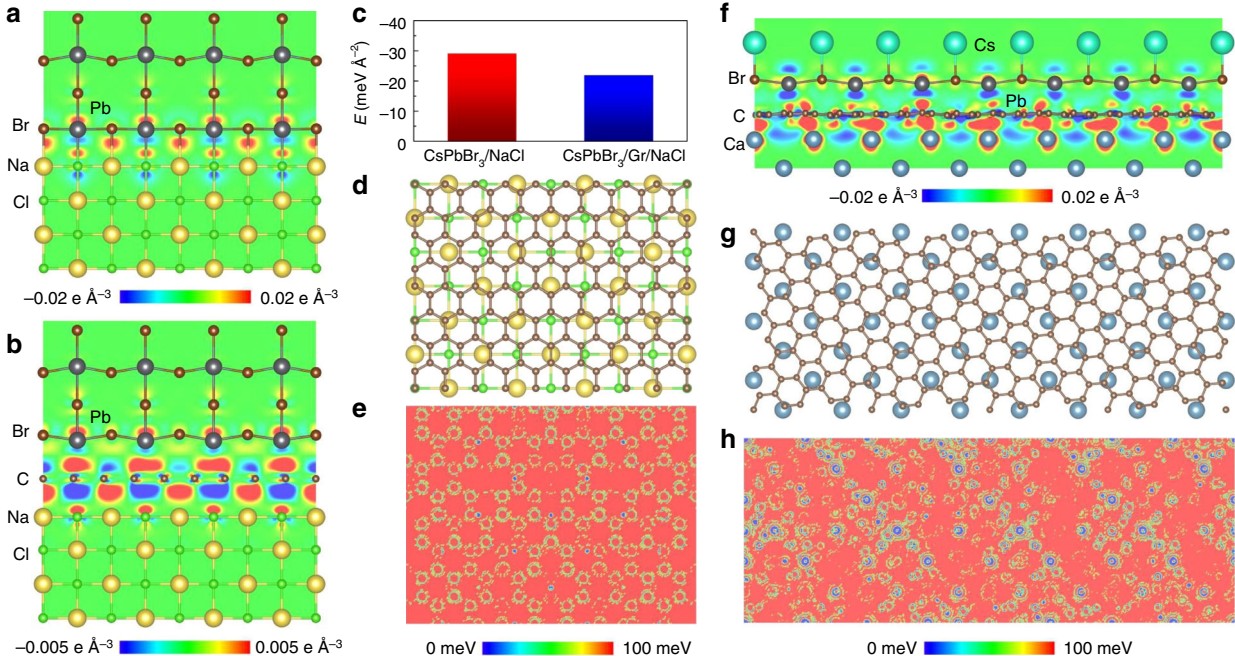

**Fig. 2** Ionic and remote atomic interactions between polar substrates and CsPbBr$_3$. **a, b** Charge transfer distributions between CsPbBr$_3$(001) and NaCl (001) (**a**), and between CsPbBr$_3$(001) and NaCl(001) with graphene intercalation (**b**). **c** Interfacial interactions between CsPbBr$_3$(001) and NaCl(001), between CsPbBr$_3$(001) and monolayer graphene buffered NaCl(001). **d** Atomic stacking between NaCl(001) top layer and buffered monolayer graphene. **e** Potential fluctuation at the epitaxial surface from NaCl(001) through monolayer graphene, where blue pattern is consistent with atomic stacking of NaCl (001) in (**d**). **f** Charge transfer distribution between CsPbBr$_3$(011) and CaF$_2$(001) with graphene intercalation. **g** Atomic stacking between CaF$_2$(001) top layer and coated monolayer graphene. **h** Potential fluctuation at the epitaxial surface from CaF$_2$(001) through monolayer graphene, where blue pattern is consistent with center site of four Ca atoms in (**g**)

the various morphologies of specific interesting areas on a typical sample, which are ionic epitaxial thin film at high supersaturation in Fig. 3d and left side of 3e, ionic epitaxial flakes at low supersaturation in Supplementary Fig. 9b, remote epitaxial thin film and flakes at high supersaturation in right side of Fig. 3e, f, respectively, remote epitaxial flakes at low supersaturation in Fig. 3g, h and Supplementary Fig. 9c. These morphologies are well consistent with the theoretical model in Fig. 3b. By counting the number of flakes at the region shown in Fig. 1e for the remote epitaxy case, with the assumption of constant nucleation rate during the growth and the growth time of 15 minutes for the sample of Fig. 3g, the nucleation rate is estimated to be around $4.4 \times 10^5 \, cm^{-2} \, s^{-1}$.

**Dislocation formation in ionic and remote epitaxy.** Dislocations are often harmful to device performance. Under thermodynamically favorable condition, dislocation-free crystals are accessible when the growth of crystals from solutions is at low supersaturation or from melts is weakly supercooled[55]. However, dislocation formation is thermodynamically favored in highly mismatched heteroepitaxy system if the film thickness is above its critical thickness (in order to relax the misfit stain energy in epilayer[56]). Traditionally, a buffer layer can be introduced into epitaxial growth to reduce dislocation density[57]. By using a special buffer layer of graphene to reduce the interfacial energy between epilayer and substrate, the epitaxial growth mechanism (thermodynamics and kinetics) in remote epitaxy is significantly different from that in traditional epitaxy. Thus, different strain relaxation mechanism is expected. Molecular-dynamics (MD) simulations are utilized to study the growth and strain relaxation mechanisms in ionic epitaxy, remote epitaxy and van der Waals epitaxy. Figure 4a, d and g shows MD simulations of the side views of ionic epitaxy, remote epitaxy and van der Waals epitaxy,

respectively. The dislocations are indicated by blue arrows, as shown in Fig. 4a–c. Supplementary Fig. 15a and Supplementary Fig. 16 show the side and top views of the formation of dislocation in ionic epitaxy, respectively. Surprisingly, no dislocation can be formed in both remote and van der Waals epitaxy. In order to release strain energy between epilayer and substrate during the growth, large interfacial energy in ionic epitaxy enables the formation of dislocations. Misfit dislocations at the interface are observed in the side view in Supplementary Fig. 15a and threading dislocations are observed in top views in Supplementary Fig. 16. For both remote and van der Waals epitaxy, small interfacial energy enables small crystal islands gliding to big ones. The gliding of crystal islands is indicated by arrows in the side view of Supplementary Fig. 15b (remote epitaxy), top views of Supplementary Fig. 17 (remote epitaxy) and Supplementary Fig. 18 (van der Waals epitaxy). Top views of Supplementary Fig. 16-18 also show the decrease of nucleation sites and the ability of gliding with decreasing interfacial energy (from ionic epitaxy, remote epitaxy to van der Waals epitaxy). Supplementary Movie 4, 5 and 6 show the side views of the entire growth processes of ionic epitaxy, remote epitaxy and van der Waals epitaxy, respectively. Supplementary Movie 7, 8 and 9 are the top views of ionic epitaxy, remote epitaxy and van der Waals epitaxy, respectively.

STEM, HRAFM, and HRTEM were utilized to characterize an as-grown remote epitaxial sample CsPbBr$_3$/Gr/NaCl. Perfect crystallite atomic structures of CsPbBr$_3$ are shown in cross-sectional images of Fig. 1k, Supplementary Fig. 10a, b, and Fig. 4j, k obtained by Cs-corrected STEM. FFTs in their insets confirm the orthorhombic phase of CsPbBr$_3$. Two ferroelastic domains with zone axes of [001] and [1$\bar{1}$0], as shown in Fig. 4j, k, respectively, are consistent with two out-of-plane peaks in the XRD $\omega$−2$\theta$ scan in Supplementary Fig. 3. Supplementary Fig. 10c

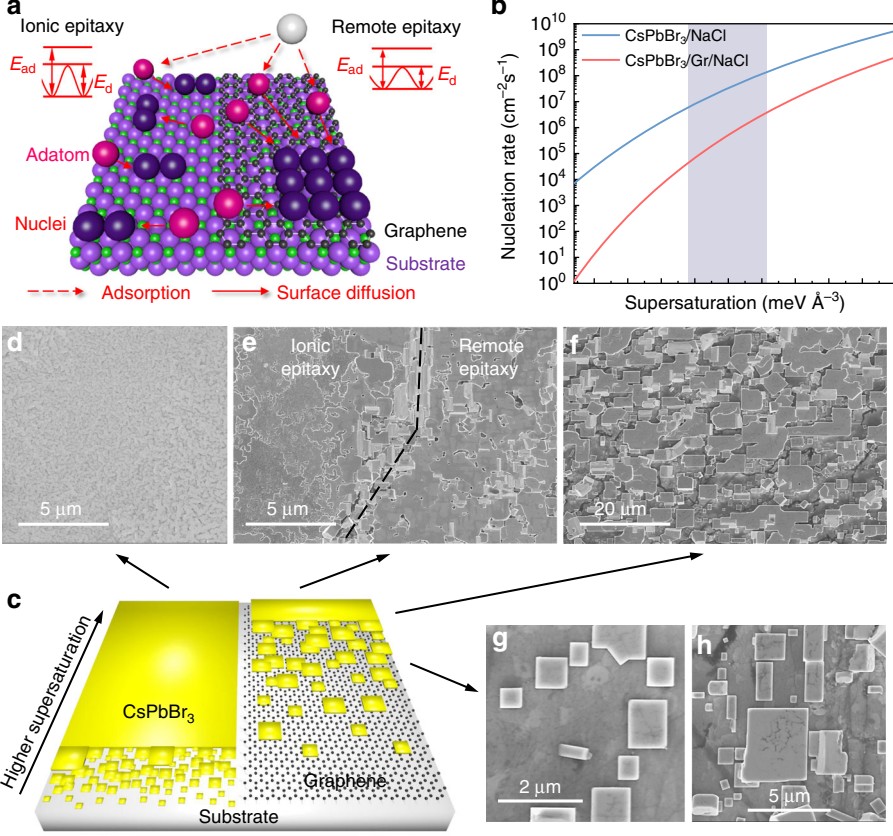

**Fig. 3** Growth kinetics of ionic and remote epitaxy. **a** Schematic illustration of atomistic nucleation process at the initial stages of both ionic and remote epitaxy. **b** Nucleation rate as a function of supersaturation for both ionic epitaxy ($CsPbBr_3$/NaCl) and remote epitaxy ($CsPbBr_3$/Gr/NaCl). **c** Schematic illustration of the nucleation process at the final stages of both ionic and remote epitaxy. **d**–**h** SEM images of specific interesting regions on a typical sample denoted by black arrow

(with zoom-in part shown in Fig. 4m) and Supplementary Fig. 10d (with zoom-in part shown in Fig. 4n) are inverse FFTs of their insets with highlighted white spots to highlight lattice fringes from Supplementary Fig. 10b, indicating no misfit dislocation. The atomic-scale image of a surface region of remote epitaxial film obtained from HRAFM exhibits perfect crystallite structure as well, as shown in Supplementary Fig. 19a. Figure 4l and Supplementary Fig. 19b are the inverse FFTs of their insets to highlight atomic structure fringes from Supplementary Fig. 10b and Supplementary Fig. 19a, indicating no threading dislocation. However, after carefully searching of many images, threading dislocations can be observed in remote epitaxial $CsPbBr_3$ from some HRAFM and HRTEM images, as shown in highlighted lattice fringes in Supplementary Fig. 19d, Fig. 4o and Supplementary Fig. 20d–f (inverse FFTs of their insets transformed from Supplementary Figs. 19c, 20c and 20a–c, respectively). These threading dislocations observed from the remote epitaxial sample might be formed during large grain coalescence stage. More threading dislocations are found at a surface region of ionic epitaxial film, as shown in Supplementary Fig. 21d (inverse FFT of its inset transformed from Supplementary Fig. 21c), while similar perfect crystallite lattice is found at another surface region, as shown in Supplementary Fig. 21b (inverse FFT of its inset transformed from Supplementary Fig. 21a). It is noted that the density of dislocations could not be estimated from these atomic-scale images due to low density and uneven distribution of dislocations.

van der Waals epitaxy of halide perovskite on a native oxide Si (100) substrate (non-polar) with transferred graphene on its surface (Gr/Si(100)) has been demonstrated in Supplementary

Fig. 22a. As expected, remote epitaxy shows better epitaxy and higher nucleation rate than van der Waals epitaxy with Gr/Si (100) as substrate. This highlights the importance of extra potential guide of substrate in remote epitaxy. To improve substrate interaction, mica is chosen to substitute the Gr/Si(100) substrate. As shown in Supplementary Fig. 20b, c, better crystal quality of halide perovskite (compared to Gr/Si) has been achieved in our growth. However, due to the in-plane symmetry mismatch between halide perovskite and mica, the epitaxial halide perovskite flakes show in-plane rotation (Supplementary Fig. 22d). Without the guide of potential field with matched symmetry from the polar substrate, the in-plane film quality in such quasi-van der Waals epitaxy is lower than that in remote epitaxy. However, the out-of-plane crystallinity of halide perovskite is better compared to that for remote epitaxy, as shown in XRD $\theta-2\theta$ scanning and rocking curve in Supplementary Fig. 20d, f, respectively.

Additionally, the interfacial energy between substrate and epilayer could control the ferroelastic phase transition kinetics. Ionic epitaxial $CsPbBr_3$ shows small spatial periodicity of the ferroelastic domains, while remote epitaxial one shows large spatial periodicity of the ferroelastic domains. Detailed results and discussion of controlling domain wavelength via remote epitaxy are presented in Supplementary Note 1 and Supplementary Figs. 26 and 27.

**Misfit and threading dislocation-carrier dynamics relation.** According to Matthew's theory, one could evaluate the lower limit of both misfit and threading dislocations densities in

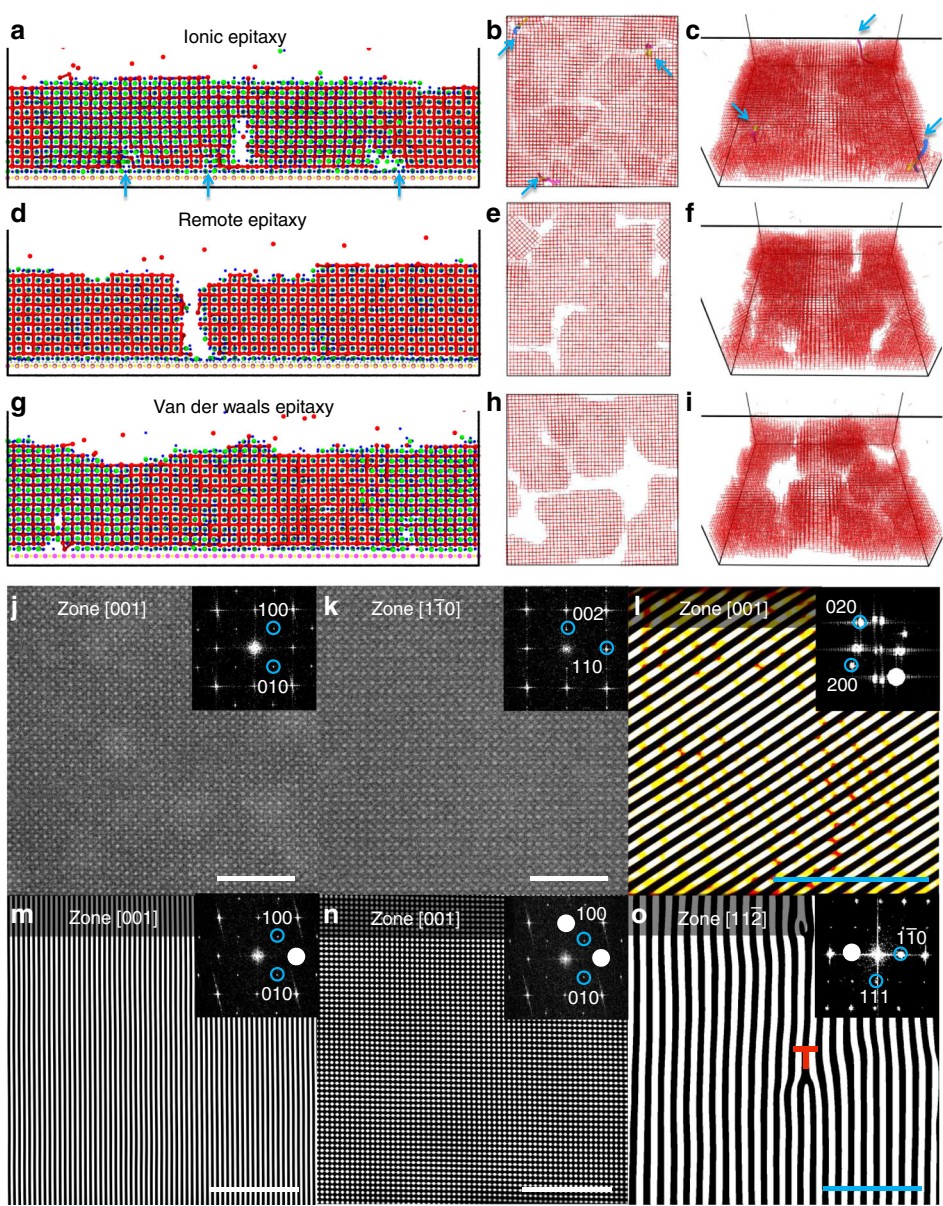

**Fig. 4** Molecular-dynamics simulations of ionic epitaxy (**a–c**), remote epitaxy (**d–f**) and vdW epitaxy (**g–i**). **a**, **d** and **g** are side views. **b**, **e** and **h** are top views. **c**, **f** and **i** are bird views. Dislocations are indicated by blue arrows. **j**, **k** STEM images for different domains of CsPbBr$_3$/Gr/NaCl with zone axes of [001] (j) and [1$\bar{1}$0] (k) and FFTs in their insets. **l** Inverse FFT of its inset transformed from the HRAFM image (Supplementary Fig. 19a) for CsPbBr$_3$/Gr/NaCl. **m**, **n**, Zoom-in inverse FFTs of their insets transformed from the STEM images Supplementary Fig. 10a, b for CsPbBr$_3$/Gr/NaCl, respectively. **o** Inverse FFT of its inset transformed from the HRTEM image (Supplementary Fig. 21c) for transferred CsPbBr$_3$. A dislocation is indicated in **o**. Scale bars of **j**, **k**, **l**, **o** are 5 nm. Scale bars of **m**, **n** are 10 nm

epitaxial film when lattice mismatch and film-substrate interaction strength are given. Clearly, remote epitaxial film is expected to take much less dislocations than ionic epitaxial film. To understand the impact of dislocations on the carrier dynamics of halide perovskite, it is natural to compare the optoelectronic properties in these two types of films. The optical properties of a typical sample have been studied by steady-state photo-luminescence (PL) measurements firstly, as shown in Fig. 5a, b. An intensive green light can be seen from the optical microscopy image in the inset of Fig. 5b, due to the band emission. Remarkable enhancements of around 3.7 and 2 times in the PL intensity can be clearly observed in both CsPbBr$_3$ thin film and flake from remote epitaxy comparing to these from ionic epitaxy, respectively.

The optical properties of the as-grown CsPbBr$_3$ thin films and flakes from both remote epitaxy and ionic epitaxy have been further investigated by time-resolved photoluminescence (TRPL) spectroscopy. The schematic illustration in Fig. 5c describes the carrier dynamics in semiconductor thin film, e.g. halide perovskite in the present case, including the effects of dislocations. The observed PL recombination dynamics could include contributions from Shockley–Read–Hall processes, radiative recombination, Auger recombination, and interface recombination[58,59]. Misfit and threading dislocations can be formed above the critical thickness of the film in semiconductors, e.g. GaP[60], GaN[4], inducing defect trap states, as shown in Fig. 5c, thereby influence or even determine the effective carrier lifetime via non-radiative recombination. By taking account of the grain

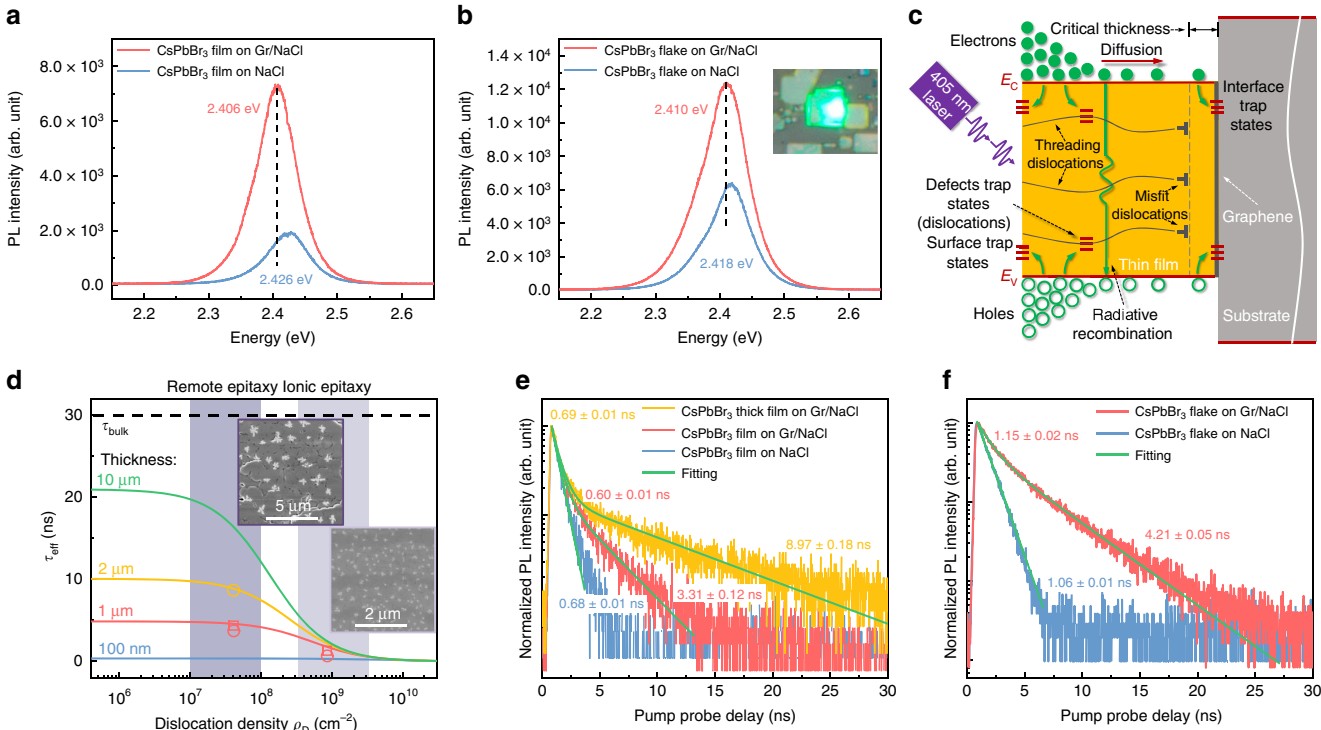

**Fig. 5** Dislocation-carrier dynamics relation in remote and ionic epitaxy. **a, b,** Steady-state PL of CsPbBr₃ films (**a**) and flakes (**b**) in both remote and ionic epitaxy. **c** Schematic illustration of carrier dynamics in CsPbBr₃. **d** Effective carrier lifetime as a function of dislocation density with different sample thickness and SEM images for ionic (right one) and remote (left one) epitaxial samples after etching. Experimental data for thin film and flake are indicated in circle and square, respectively. Remote epitaxy and ionic epitaxy regions painted in dark purple and light purple, respectively. **e, f,** TRPL of CsPbBr₃ films (**e**) and flakes (**f**) in both remote and ionic epitaxy

boundary recombination and dislocation recombination, with the assumption of same diffusivity of the excess carriers as the minority carriers at low injection and no junction on an array of dislocations in an infinite semiconductor, the effective lifetime can be modified from the previous works by refs. [28,54,61,62]

$$\frac{1}{\tau_{eff}} = \frac{1}{\tau_{bulk}} + \frac{1}{\tau_{int}} + \frac{1}{\tau_{gb}} + \frac{1}{\tau_{dis}} = \frac{1}{\tau_{bulk}}$$
$$+ \frac{1}{\frac{t}{2S_{int}} + \frac{1}{D}\left(\frac{t}{\pi}\right)^2} + \frac{1}{\frac{d}{2S_{gb}} + \frac{1}{D}\left(\frac{d}{\pi}\right)^2} + \frac{4\pi D\rho_D}{-ln\left(\pi\rho_D r_0^2\right) - \frac{6}{5}} \quad (5)$$

where $\tau_{gb}$ is the grain boundary carrier lifetime, $\tau_{dis}$ is the dislocation carrier lifetime, $S_{int}$ and $S_{gb}$ is the recombination velocity at the interface and grain boundary, respectively, $d$ is the lateral grain size, $t$ is the vertical thickness, $D$ is the diffusivity, $\rho_D$ is the threading dislocation density, and $r_0$ is the effective core radius of a dislocation inside which the lifetime is zero. The recombination velocity at interface $S_{int}$ is influenced by the growth condition and interfacial chemistry, while the recombination velocity at grain boundary $S_{gb}$ is fixed for the same material. Here we assume the recombination velocity at both interface and grain boundary are the same ($1.5 \times 10^4$ cm s⁻¹ was taken[63]) and focus on the dislocation recombination effect. The diffusivity $D$ is taken to be 0.35 cm² s⁻¹[63], $r_0$ is taken to be $5.87 \times 10^{-8}$ cm (the lattice constant of cubic CsPbBr₃). The lateral grain size $d$ of around 1–5 μm was roughly measured by the SEM images Fig. 3. By using equation (5) with constant values of sample thickness and grain size, the effective lifetime of halide perovskite CsPbBr₃ is plotted as a function of dislocation density, as shown in Fig. 5d and Supplementary Fig. 23 for different sample thickness and grain size, respectively. Thus, the effect of dislocation density on

the effective lifetime of halide perovskite CsPbBr₃ is quantitatively analyzed. The thickness and the grain size can influence the effective lifetime as well, as shown in lines with different colors in Fig. 5d and Supplementary Fig. 23, respectively. Obviously, the effective lifetime reduces drastically with increasing dislocation density if we assume dislocations in halide perovskite provide recombination centers.

To experimental show whether the dislocations contribute significant recombination centers or not, we have further quantified the dislocation density and correlated them with the TRPL lifetime. The threading dislocation density can be estimated by dislocation etch pits. The etching technique for CsPbBr₃ is similar to that for NaCl[64], but with different concentration of etching solution. Dislocation etch rosettes have been formed on the surface of the remote epitaxial sample, ionic epitaxial sample and NaCl substrate for comparison, as shown in the left and right inset of Fig. 5d and Supplementary Fig. 5d. From the density of dislocation etch rosettes, for the ionic epitaxy, the threading dislocation density is estimated to be around $9 \times 10^8$ cm⁻², while for the remote epitaxy, it is remarkably reduced to around $3 \times 10^7$ cm⁻². Both remote epitaxy and ionic epitaxy regions are painted in dark gray and light gray in Fig. 5d, respectively.

The effective lifetimes have been extracted from the measured TRPL spectra for both CsPbBr₃ thin film and flake in both remote and ionic epitaxy in Fig. 5e, f, respectively. Two kinds of recombination regimes can be extracted, including the short carrier lifetime dominated by the interface recombination and the long carrier lifetime dominated by the bulk recombination. The grain boundary and dislocation recombination can affect both regimes' carrier lifetime. For the remote epitaxial thin film comparing to the ionic epitaxial one, the effective lifetime of long-life carrier is greatly increased from 0.68 to 3.31 ns, while for the

remote epitaxial flake it increases from 1.06 to 4.21 ns. The significant prolonged carrier lifetime of about four times in magnitude could be attributed to the more than one order of magnitude decrease in threading dislocation density. With doubled growth time, thick film shows effective lifetime up to 8.97 ns, as shown in Fig. 5e. In single flakes with much smaller lateral size comparing to the thin film, the carrier lifetimes, as shown in Fig. 5f, however, are slightly longer than these in films, as shown in Fig. 5e. The extracted carrier lifetimes are good consistent with the hypothesis in Fig. 5d, indicating the predominant factor of dislocation recombination for the carrier dynamics in halide perovskite $CsPbBr_3$ and the remarkable improvement of carrier lifetime in remote epitaxy. Similar enhancement of the effective lifetime is also observed in the case of $CsPbBr_3$ grown on $Gr/CaF_2$, as shown in Supplementary Fig. 24. These evidence clearly show that threading dislocation is as important as grain boundary or point defect in halide perovskite for physical properties and device performance.

Two gold stripe contacts are deposited onto the $CsPbBr_3$ thin film with a physical mask on top by e-beam evaporation and a simple device of photodetector ($Au/CsPbBr_3$(remote epitaxial thin film)/Au) is fabricated, as shown in Supplementary Fig. 25a. The photo responses under UV light (405 nm laser, around 0.5 mW) are measured for both devices made by ionic and remote epitaxial thin films, as shown in Supplementary Fig. 25b. Supplementary Fig. 25c, d are the enlarged rising and falling parts in Supplementary Fig. 25b, respectively, revealing a faster rise time in remote epitaxial thin film-based device (Supplementary Fig. 25c) and an additional decay tail with a long decay time of around 8.08 s in ionic epitaxial thin film-based device (Supplementary Fig. 25d). These observations, i.e. fast rise time and disappearance of longer decay tail in remote epitaxial thin film-based device, could be attributed to the low density of dislocations in remote epitaxial thin film. Similar observations can be found in other photodetectors, such as AlGaN[65].

## Discussion

Epitaxial halide perovskite has been synthesized via a remote epitaxy approach. DFT simulations and experimental implementation show that the modified electrostatic potential of polar substrates coated with graphene enables tunable film-substrate interaction strength leading to controlled dislocation density in epilayer. The epitaxial relationships have been revealed by high resolution XRD RSMs and pole figures, which are out-of-plane $CsPbBr_3(001)\|Gr/NaCl(001)$ and in-plane $CsPbBr_3[100]\|Gr/NaCl[100]$ for remote epitaxy of $CsPbBr_3$ on Gr/NaCl and out-of-plane $CsPbBr_3(011)\|CaF_2(001)$ and in-plane $CsPbBr_3[100]\|CaF_2[010]$ or $CsPbBr_3[0\bar{1}1]\|CaF_2[010]$ for remote epitaxy of $CsPbBr_3$ on $Gr/CaF_2$. The weak film-substrate coupling in remote epitaxy suppresses the nucleation and promotes the growth, and a semi-quantitative epitaxial nucleation and growth model is proposed to understand the growth kinetics. Further, the ferroelastic phase transition kinetics can be controlled by film-substrate coupling and remote epitaxy yields large spatial periodicity of the ferroelastic domains in $CsPbBr_3$. In the cases of remote and van der Waals epitaxy with low interfacial energy, the dislocation-free or low-density dislocation mechanism is demonstrated by molecular-dynamics simulations. HRTEM, HRAFM and Cs-corrected STEM studies show the atomic and lattice structure of the remote epitaxial films with very low-density threading dislocations. The controlling of dislocation (threading) density, which is estimated from etch rosettes, enables the unveiling of the dislocation-carrier dynamic relation in halide perovskite. Comparing to ionic epitaxy, remote epitaxy improves the PL intensity by around 3 times and the carrier lifetime by around 4 times in magnitude, indicating that the dislocation recombination is one of

the predominant components for the carrier recombination mechanisms of halide perovskite. Our study reveals dislocation as another critical defect on influencing halide perovskite's optoelectronic properties in addition to grain boundary, point defects and phase impurity, and calls for further investigations on their control to tune halide perovskite device performances.

## Methods

**Graphene transfer**. The graphene used in this work was purchased from Graphene Laboratories Inc. (Calverton, New York, US). The graphene was synthesized on Cu foils by chemical vapor deposition. For transfer of graphene, poly(methyl methacrylate) (PMMA) was spin-coated and baked on graphene/Cu to protect the graphene on the front side of Cu foils. The rear side of Cu foils was then treated in $O_2$ plasma etcher to remove unwanted graphene. Afterward, the PMMA/graphene/Cu stack was placed in ammonium persulfate aqueous solution (60 g L$^{-1}$) to etch Cu. Once Cu was etched, the PMMA/graphene was rinsed in water (isopropyl alcohol) several times and then scooped out using the $CaF_2$ (NaCl) substrate. The PMMA/graphene/$CaF_2$(NaCl) was dried in air. In the end, PMMA was dissolved in acetone.

**Remote epitaxy and ionic epitaxy of $CsPbBr_3$ on NaCl and $CaF_2$ substrates**. Cesium bromide (CsBr, 99%, Sigma-Aldrich) was placed in the furnace heating center and lead(II) bromide ($PbBr_2$, 99%, Sigma-Aldrich) was placed about 10 cm away from the cesium bromide. The heating temperature was controlled at about 500 °C. Argon flow of 200 sccm was introduced to the system to maintain the pressure at about 1.4 Torr during the growth. The growth process lasted for 15 min. The details can be found in our previous report[54]. Prior to a growth, the monolayer graphene was transferred onto the substrates. The details on the transfer process can be found in our previous report[66].

**Dislocation etching technique**. A solution of $FeCl_3$ in glacial acetic acid was used for as-grown $CsPbBr_3$ and substrate NaCl for comparison. Dislocation etch rosettes were obtained for both NaCl and $CsPbBr_3$ when they were moderately agitated for 60 s in $2.5 \times 10^{-2}$ mol L$^{-1}$ and $2.5 \times 10^{-4}$ mol L$^{-1}$ $FeCl_3$ in glacial acetic acid, respectively. After that, the samples were rinsed in acetone and dried in compressed air.

**Focused ion beam (FIB)**. The lamellar specimens for HRTEM observation were prepared by a dual-beam scanning electron microscopy (SEM)/focused ion beam (FIB) system (FEI Quanta 3D FEG and FEI Helios) based on site-specific in situ lift-out technique. Before the ion thinning, a metallic protective layer of Cr with a thickness between 30 and 40 nm was first deposited on the sample surface. Then, a platinum supporting layer with a thickness between 1 and 1.5 μm was further deposited on the targeted area using the ion beam deposition inside the chamber. This dual-beam system is capable of fast milling at a high voltage of 30 kV and subtle thinning at the final stage using a few kV (2–5 kV) Ga$^+$ ion milling. During ion thinning, the sample was tilted between 1 and 2° towards the ion beam in order to avoid significant damage to the sample structure. The ion beam current was between 10 and 15 pA.

**X-ray diffraction Characterization**. X-ray diffraction was done with a Panalytical X'Pert PRO MPD system with a Cu Kα source and a hybrid mirror with a two-bounce two-crystal Ge(220) monochromator, yielding a parallel incident beam with a wavelength $\lambda_{K\alpha1} = 1.5406$ Å, a divergence of 0.0068°, and a width of 0.3 mm. Sample alignment included height adjustment as well as correction of the $\omega$ and $\chi$ tilt angles by maximizing the substrate peak intensity. Short-range symmetric $\omega$$-2\theta$ scans were obtained using a 0.04 radian Soller slit in front of a PIXcel solid-state line detector operated in receiving mode with a 0.165 mm active length, corresponding to a $2\theta$ opening of less than 0.04°. $\omega$-rocking curves were obtained using constant $2\theta$ angles corresponding to $CsPbBr_3$ 004 reflections (indexed with tetragonal structure $P4/mbm$) and using the same parallel beam geometry as used for $\omega$$-2\theta$ scans. Asymmetric high-resolution reciprocal space maps (HR RSM) around 224 reflections (indexed with pseudocubic to simplify the analysis) were obtained using a line detector in scanning mode, operating all 255 channels. This is done using a small angle (~20°) between the sample surface and the diffracted beam to cause a beam narrowing which increases the $2\theta$ resolution and therefore facilitates fast high-resolution reciprocal space mapping. Then the data was shown as color filled iso-intensity contour maps in a logarithmic scale, plotted within $k$-space where $k_\perp = 2\sin\theta\cos(\omega-\theta)/\lambda$ and $k_\parallel = 2\sin\theta\sin(\omega-\theta)/\lambda$ correspond to directions perpendicular and parallel to the substrate surface along perpendicular [001] and [110] directions, respectively. XRD pole figures were obtained using a point focus optics with a poly-capillary x-ray lens that provides a quasi-parallel Cu Kα beam with a divergence of less than 0.3° to minimize defocusing effects associated with the non-uniform sample height due to the sample tilt. In addition, $\omega$-$2\theta$ scans with a divergent beam Bragg-Brentano geometry were acquired over a large $2\theta$ range from 5-90° in order to detect small inclusions of possible secondary phases or misoriented grains.

**Microscopy characterization.** Morphology of the halide perovskite thin films and flakes was characterized by a Nikon Eclipse Ti-S inverted optical microscope. Scanning Electron Microscope (SEM) FEI Versa 3D was used for surface morphology and thickness (cross-sectional SEM) analysis. The high resolution TEM images and electron diffraction pattern for transferred $CsPbBr_3$ flakes were collected with FEI F20 TEM operated at 200 kV. The cross-sectional interfacial morphologies and crystallite structures were evaluated by high resolution transmission electron microscope (HRTEM) JEM2010 operating at 120 kV, JEOL 2100 F operating at 200 kV and scanning transmission electron microscope (STEM) FEI Titan G2 80-200 CREWLEY operating at 80 kV.

**High resolution atomic force microscopy (HRAFM).** The surface morphologies were investigated by high resolution atomic force microscopy (AFM, Cypher ES, Oxford Instruments) in ambient air. To derive the atomic-scale structures of the crystallite surface, a sharp $Si_3N_4$ probe (TR400PB, Olympus) was used in the contact mode. The lateral force signal was recorded in order to atomically resolve the recorded image as the friction force is sensitive to the evolution of surface crystal lattice. To avoid surface damage, the applied normal load was controlled in the range of 10-20 nN and the sliding velocity was set at 10-15 Hz. The recorded image was processed by the Asylum Research Igor Pro Software.

**Piezoelectric force microscopy (PFM).** The piezoelectrics properties of the samples were measured by piezoelectric force microscopy (PFM) in an AFM system (MFP 3D, Oxford Instruments). A conductive probe with platinum overall coating (ElectriMulti75-G, BudgetSensors) was used, with a drive voltage up to 10 V applied to the conductive tip. When the tip is brought in contact with the sample surface, the local piezoelectric response can be detected by recording the distortional motion of the cantilever. Thus the piezoresponse can be estimated measuring the vibrating amplitude of the cantilever per unit drive voltage. The scanning velocity was 1 Hz. Both the amplitude and phase signals of the piezoresponse within a specific domain were acquired.

**Raman spectroscopy.** Confocal Raman spectroscopy was performed using a Horiba XploRA Nano system. One of two excitation wavelengths, 532 and 638 nm, was utilized based on the needs of different situations. A typical Raman spectrum was collected under following conditions: 100X objective lens, 3 s integration time, 10 acquisitions, 1200 grooves/mm grating, 4 mW laser power.

**Photoluminescence (PL) characterization.** The PL of samples were measured via a customized PL system consisting of a Picoquant 405 nm pulsed laser with a 2 mW power and a repetition of 4 MHz, a Thorlabs 4 Megapixel Monochrome Scientific CCD Camera, a Princeton Instruments SP-2358 spectrograph, and the same optical microscope that focuses the laser via a ×50 objective lens.

**Time resolved photoluminescence (TRPL) characterization.** Time resolved PL was measured in the same PL system. A Picoquant PDM series single-photon detector synchronized with the pulsed laser source was used to collect the time domain PL information.

**DFT calculation.** The DFT calculations were performed with the Vienna ab initio Simulation Package (VASP)[67]. The core electrons were described by the projector-augmented-wave (PAW) method[68] and the electron exchange and correlation were modeled within the generalized gradient approximation (GGA) using the Perdew-Burke-Ernzerhof (PBE) form[69]. The non-local optB86b-vdW exchange-correlation functional was used to describe the dispersion interaction (vdW forces) approximately[70], as its accurate descriptions of geometrical structures and energies. The plane wave basis kinetic energy cut off was set to 400 eV.

For $CsPbBr_3(001)$ growth on Gr/NaCl(001), the calculation supercells were constructed as four layers of $CsPbBr_3(001)$ ($a = 5.89$ Å, cubic phase is chosen for calculation) sitting on four layers of NaCl(001) ($a = 5.57$ Å), with about 5% compressive strain imposing on $CsPbBr_3(001)$. The remote atomic interaction of NaCl(001) though monolayer graphene was simulated with $4 \times 3$ $CsPbBr_3$ sitting on monolayer graphene-coated $4 \times 3$ NaCl(001). The selection of this model is mainly considering to decrease strain on graphene intercalation. The top NaCl (001) layer and atoms above were allowed to relax until the forces on all the relaxed atoms were less than 0.05 eV/Å. The vacuum gaps in simulation supercells are larger than 10 Å.

For $CsPbBr_3(011)$ growth on Gr/CaF$_2$(001), the calculation supercell was constructed as three layers of $CsPbBr_3(011)$ ($a = 5.89$ Å, $b = 8.33$ Å) sitting on three layers of CaF$_2$(001) ($a = 5.35$ Å). The remote atomic interaction of CaF$_2$(001) through monolayer graphene was simulated with $7a \times 2b$ $CsPbBr_3(011)$ sitting on monolayer graphene-coated $8 \times 3$ CaF$_2$(001). The selection of this model is mainly based on the minimization of the mismatch between $CsPbBr_3$ and CaF$_2$(001). For the epitaxial relation of $CsPbBr_3(011)\|CaF_2(001)$, the mismatch is 3.8% along $CsPbBr_3[100]\|CaF_2[100]$ and $-3.7\%$ along $CsPbBr_3[0\bar{1}1]\|CaF_2[010]$, which is remarkably smaller than the mismatch of $-9\%$ along $CsPbBr_3[100]\|CaF_2[100]$ for $CsPbBr_3(001)\|CaF_2(001)$. The strain imposed on graphene is less than 2%. The top CaF$_2$(001) layer and atoms above were allowed to relax until the forces on all the relaxed atoms were less than 0.05 eV/Å. The vacuum gap in simulation supercell is larger than 20 Å.

**Molecular dynamic simulation.** We designed a minimalist's molecular-level model to simulate the deposition and subsequent epitaxial growth of a model perovskite film on various substrates. This model enables us to investigate structural features of a three-component epitaxial film such as misfit dislocations with moderate computational resources. In addition, observations made with such a generic perovskite model are likely to be valid for perovskite systems in general, as opposed to observations stemming from unique features of a particular material.

There are three particle species (A, B, and C) in the model perovskite crystal $ABC_3$. The particle-particle interaction is described by a truncated Lennard-Jones (LJ) potential:

$$\phi_{LJ}(r) = 4\varepsilon_{\alpha\beta}\left(\frac{\sigma_{\alpha\beta}^{12}}{r^{12}} - \frac{\sigma_{\alpha\beta}^{6}}{r^6}\right) - 4\varepsilon_{\alpha\beta}\left(\frac{\sigma_{\alpha\beta}^{12}}{r_{\alpha\beta,c}^{12}} - \frac{\sigma_{\alpha\beta}^{6}}{r_{\alpha\beta,c}^{6}}\right) \quad (6)$$

We map the generic $ABC_3$ perovskite to $CsPbBr_3$ to obtain the mass and size of each particle. It should be noted that we do not intend to use LJ potential to describe the complex interactions including long-range Coulombic force within $CsPbBr_3$. Yet, even LJ potential, with the proper choice of bond strengths together with the bond length in $CsPbBr_3$, leads to the crystallization of perovskite. The pair-dependent bond strength, bond length, and cutoff are listed in Supplementary Table 1. The energy and length units are chosen as $\varepsilon_{BC}$ and $\sigma_{BC}$ for convenience. Given a unit mass of $m_0$, the particle mass for A, B and C are 1.66, 2.6 and 1.0 $m_0$, respectively.

Consequently, the temperature unit is $\frac{\varepsilon_{BC}}{k_B}$ ($k_B$ is the Boltzmann constant), and the time unit is $\tau_0 = \sigma_{BC}\sqrt{\frac{m_0}{\varepsilon_{BC}}}$. The melting point of the $ABC_3$ perovskite was estimated to be about 0.13 $\frac{\varepsilon_{BC}}{k_B}$. Therefore, the reduced units above for this model perovskite system can be written in real units by mapping to $CsPbBr_3$ as, $\varepsilon_{BC} = 0.557$ eV (using the melting point of $CsPbBr_3$ as 567 °C), $\sigma_{BC} = 0.249$ nm (using the lattice constant of $CsPbBr_3$ as 0.587 nm), $m_0 = 79.9$ amu (using the mass of Br), and $\tau_0 = 0.3$ ps.

The perovskite potential parameters and the rational are discussed as follows. Only B–C and A–C interactions have attractive regime, while all other pair interactions are repulsive-only (the cutoffs are set at the bottom of the LJ potential well). B–C pairs have strong affinity, together with the C–C repulsion (and the proper hardsphere size), lead to BC6 octahedral formation. A-C pairs have weaker attraction than B–C pairs to favor $ABC_3$ perovskite formation rather than B–C compound formation. The lattice constant of the ideal perovskite structure is about 2.2449 $\sigma_{BC}$ at zero temperature, and around 2.36 $\sigma_{BC}$ at the deposition temperature of 0.1 $\varepsilon_{BC}/k_B$. The temperature is controlled by Nose-Hoover thermostat. The equations of motion are numerically integrated using the velocity-Verlet algorithm with a time step of 0.0056 $\tau_0$.

We have investigated three different types of substrates for the deposition of perovskite $ABC_3$. The first type of substrate (termed Substrate I) has a strong affinity with the deposited film, which is analogous to 'ionic epitaxy'. There are two types of particles (D and E particles) forming a rigid substrate. E particles form a square lattice with a lattice constant of 1.1352 $\sigma_{BC}$. One quarter of the E particles were replaced by D particles such that D particles themselves form a square lattice with a lattice constant of 2.2704 $\sigma_{BC}$. Due to the strong affinity between C and D particles, epitaxy growth is expected with a lattice mismatch of about 3.8% at the deposition temperature. The second type of substrate (termed Substrate II) is identical to Substrate I with an additional model 2D material (graphene-like) layer acting as a spacer so as to weaken the substrate-film affinity, which is analogous to 'remote epitaxy'. The model 2D material has a honeycomb lattice made of F particles, with atomic spacing of 0.971 $\sigma_{BC}$, which is 0.925 $\sigma_{BC}$ away from the original substrate. There is a very weak attraction between C and F particles to facilitate deposition. The third type of substrate (termed Substrate III) has a weak affinity to the film (identical to Substrate I except the C–D interaction is only half as strong), which is analogous to 'van der Waals epitaxy'.

The simulation system for deposition is around 113.5 $\sigma_{BC}$ by 113.5 $\sigma_{BC}$ by 90.8 $\sigma_{BC}$ for simulations with Substrate I and III, and 106.7 $\sigma_{BC}$ by 113.5 $\sigma_{BC}$ by 90.8 $\sigma_{BC}$ for simulations with Substrate II. During deposition, all atoms belong to the substrate are frozen for simplicity. For all deposition simulations, particles A, B and C are created randomly without overlapping with any existing atoms at the top of the simulation box. On average, particles A and B are created every 600 timesteps (or 3.36 $\tau_0$), while particles C are created every 1.12 $\tau_0$ to maintain the correct stoichiometry. All particles are given an initial velocity towards the substrate. The deposition temperature is set to be 0.1 $\varepsilon_{BC}/k_B$ (about 77% of the melting point).

## Data availability

All the relevant data are available from the corresponding authors upon reasonable request.

## Code availability

All the relevant code are available from the corresponding authors upon reasonable request.

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

## Acknowledgements

We acknowledge the financial support from the National Key Research and Development Program of China (Grant No. 2017YFB0702100), National Natural Science Foundation of China (Grant No. 51705017, U1706221, and 51727901). The experimental work is supported by the NYSTAR Focus Center at Rensselaer Polytechnic Institute (RPI), C150117, NSF award under No. 1635520, No. 1712752, No. 1508410 and No. 1706815, Air Force Office of Scientific Research under award number FA9550-18-1-0116, the Office of Naval Research under award number N000141812408 and NSF MIP PARA-DIM under award number 1539918. Research performed at Brown University was supported by the National Science Foundation (OIA-1538893) and the Office for Naval Research (N00014-17-1-2232). K.B. and H.T. are grateful to the National Science Foundation (EFRI-1433311), the Center for Computational Innovations (CCI) at Rensselaer Polytechnic Institute, and the Extreme Science and Engineering Discovery Environment (XSEDE, project TG-DMR17008), which is supported by National Science Foundation grant number ACI-1053575.

## Author contributions

J.S. conceived the idea. J.S., J.F. and L.G. supervised the project. J.J., X.S. and J.S. designed the experiments. X.S. processed and prepared substrates coated with graphene. J.J. prepared flakes/thin films and performed SEM. J.J. and Z.C. performed PL measurements. J.J., Z.C. and Y.H. performed device fabrication and photo response measurements. X.S., K.B. and H.T. performed Raman spectroscopy measurements and analysis. B.W. performed XRD. J.J. and Y.H. analyzed the domain pattern. L.Z. performed AFM. Y.G. performed HRTEM at Cornell University. F.Z. and L.J. performed STEM. J.J., J.S., M.C., Z.M., Y.Z. and N.P.P. performed FIB and TEM at Brown University. X.C. performed FIB, HRAFM and PFM at Tsinghua University. Y.M. and L.G. performed DFT calculations. Y.S. performed MD simulations. J.J. analyzed data and wrote the paper. All the authors were involved in the discussion for data analysis. D.G., X.S., T.L., E.W. and J.S. revised the manuscript.

## Additional information

**Competing interests:** The authors declare no competing interests.

