## [Peer Review File · Nature Communications]

Reviewers' comments:

Reviewer #1 (Remarks to the Author):

The paper reports several aspect in respect to epitaxial CsPbBr₃ perovskites. The recently developed epitaxial growth on graphene is adopted here to control the dislocation density in the perovskite. The dislocation density is further shown to impact the carrier lifetime, which is not a surprise, but was justified to be shown in view of the discussions about "defect tolerance" in this material. A further aspect discussed in this paper is the presence of ferroelastic domains. This is a new topic in halide based perovskites and is discussed here for the CsPbBr₃ for the first time. Even though this is an interesting aspect, this part in the manuscript I find gratuitous. The story is not related to the title of the manuscript, the experimental finding are not sufficiently explained and the conclusions are not convincing. Any experimental proof of the presence of the ferroelasticity as is presented in Refs. 66-69 is totally missing. Thus, I suggest to remove everything in relation to ferroelasticity or ferroelectricity from the manuscript. I find it also unnecessary to highlight several times in the paper that something was done for the first time

Reviewer #2 (Remarks to the Author):

Recommendation: Reject, but indicate to the authors that further work might justify a resubmission

The authors made halide perovskite thin films using the remote epitaxy approaches, investigated the several physical properties. The growth mechanism of halide perovskites was discussed in terms of the electrostatic interaction between the polar substrate and perovskite. With aid of the DFT calculations and a semi-quantitative modelling, flake-like formation of halide perovskites on graphene was explained. Later the authors argued that the striped ferroelastic domains appear wider in the remote epitaxy, explained by the continuum modelling. They concluded that the dislocations in halide perovskites are strong recombination centres based on the PL measurements.

Authors report some impressive findings, but I am worried that the conclusions were not so interesting and appealing to researchers of the halide perovskites. Also, some of their conclusions seem not strongly supported by the measurements. For instance, it is concluded that dislocations recombine the carriers strongly, based on the stronger PL strength of CsPbBr₃ on graphene. But it might be caused by the larger grain size, different thickness, or different surface orientations. Moreover, the thickness of the samples is not explicitly discussed in the manuscript. Therefore, I do not recommend this manuscript to be published in Nature Communications.

(1) Why two samples, CsPbBr₃/Gr/NaCl and CsPbBr₃/Gr/CaF₂ exhibit different growth patterns? Also, it is not crystal clear why CsPbBr₃/Graphen/CaFe₂ samples were not investigated further.

(3) $\Delta\mu$ seems not well defined in the manuscript.

(4) The derivation of equation 6 might not clear to the readers. Authors might explain the derivation in the supporting information.

(5) In figure 4, the dimension of the y-axis is missing.

(6) In figure S6, the dimension of the y-axis is missing.

Reviewer #3 (Remarks to the Author):

This manuscript studies impact of remote epitaxy on the performance of halide perovskite. Since the discovery of remote epitaxy, it is the first work showing the improvement of performance in perovskite by performing remote epitaxy. The study must be meaningful for epitaxy community as well as halide-perovskite community. However, there are issues that authors must address before being considered for publication. Once all issues are completely address, the manuscript can be recommended for the publication as it contains very interesting physics and application potential.

1. In the manuscript, main argument is about dislocation. However, there is no clear evidence showing dislocations. Instead it mainly shows electrical data. Authors should change their title. I suggest "Carrier lifetime enhancement in halide perovskite via remote epitaxy" to be considered. Title along this line fits better also because the biggest advantage from remote epitaxy in halide perovskite is an enhancement of carrier lifetime, which is very meaningful for halide perovskite community.
2. There is no clear explanation regarding figure 5d. Authors need to clarify how the curve was obtained. In the manuscript, there are only two different dislocation density cases.
3. Authors need to measure AFM on the surface of both ionic-epitaxially grown one and remote-epitaxially grown one to show how good morphology they have.
4. Since graphene's properties can critical for remote epitaxy, authors should clearly reveal the properties of graphene such as crystallinity and uniformity. For example, laser confocal microscopy could be useful to confirm the uniformity of 2D materials.
5. After exfoliating epilayers from graphene, authors must show the quality of graphene, morphology of substrate, exfoliation yield, and condition of epilayers. This confirmation is important to make an argument of remote epitaxy on graphene. Thorough investigation using AFM, SEM, and TEM is needed.
6. Authors must discuss about the full detail of the process including method of graphene transfer on the substrate and condition of substrate after the transfer since they can critically impact on the quality of remote epitaxial layers. Characterization data in each process can be displayed in the supplementary. Also it would be great to show quality of epilayer depending on the transfer methods.
7. What could happen if interface between the substrate and graphene is contaminated? It could be interesting if authors include the experimental result to show some interface condition.
7. It is hard to know nucleation rate in case of ionic epitaxy through data in the manuscript. Authors need to include some SEM images of the ionic epitaxy case like figure 1e/g
8. Detail study in the CaF₂ case could be meaningful for obtaining general idea for remote heteroepitaxy. DFT calculation, TRPL study can strengthen significantly.
9. Epitaxial surface does not seem to be atomically flat. How about the homogeneity in terms of carrier transportation? Does the remote-epitaxially grown one have a better homogeneity? Conducting AFM mapping could be useful to figure out this issue.
10. Actually, dislocation is just one type of defects. Authors claim that all effect comes from dislocation reduction. If so, authors have to study more to show the effect from dislocation distinguished from other types of defects.

11. TEM at the epitaxial interface is highly recommended
12. Is there any usefulness when ferroelastic domain pattern changes? This can be very interesting.
13. In addition to lifetime measurement, device demonstration can strengthen the paper significantly.
14. STM could be powerful to observe the potential influence coming from NaCl or CaF₂.
15. English correction is needed from a native speaker.

Reviewer #1 (Remarks to the Author):

The paper reports several aspect in respect to epitaxial CsPbBr₃ perovskites. The recently developed epitaxial growth on graphene is adopted here to control the dislocation density in the perovskite. The dislocation density is further shown to impact the carrier lifetime, which is not a surprise, but was justified to be shown in view of the discussions about “defect tolerance” in this material. A further aspect discussed in this paper is the presence of ferroelastic domains. This is a new topic in halide based perovskites and is discussed here for the CsPbBr₃ for the first time. Even though this is an interesting aspect, this part in the manuscript I find gratuitous. The story is not related to the title of the manuscript, the experimental finding are not sufficiently explained and the conclusions are not convincing. Any experimental proof of the presence of the ferroelasticity as is presented in Refs. 66-69 is totally missing. Thus, I suggest to remove everything in relation to ferroelasticity or ferroelectricity from the manuscript. I find it also unnecessary to highlight several times in the paper that something was done for the first time

Our response:

We thank the reviewer #1 for carefully reading the manuscript and the insightful comments on the aspect of the discussion on ferroelastic domains. Remote epitaxy shows the ability to control the ferroelastic domain pattern in halide perovskite. We agree that although this aspect is interesting, it is not directly related to the title of the manuscript. The fundamental physics responsible for extraordinary optoelectronic performance of halide perovskite is still under debate. One explanation may be associated the subtle domain physics^{2, 3}. The domain (ferroelectric or ferroelastic domain) engineering might be a promising direction for the enhancement of the performance of halide perovskite solar cells. Our finding on the controlling ferroelastic domain via remote epitaxy might bring new inspiration to the halide perovskite community. We agree with the reviewer that this is not the key point of this manuscript and hence, we would like to move this part to supplementary information with new data added.

According to the references²⁻⁶, we have performed PFM measurements to study the domains in CsPbBr₃. The PFM amplitude and phase mapping for a local area on the remote epitaxial thin film are shown in Fig. R1a and b (Fig. S26e and f in revised SI), respectively. The contrast difference in both Fig. R1a and b indicates ferroelastic domains in CsPbBr₃. In addition, we observed two ferroelastic domains of CsPbBr₃ with two different zone axes in the cross-sectional images of CsPbBr₃/Gr/NaCl evaluated by low-voltage Cs-corrected scanning transmission electron microscope (STEM), as shown in Fig. R2a and b (Fig. 4j and k in revised main text).

Fig. R1 PFM amplitude (a) and phase (b) mapping of the piezoresponse for a local area on a remote epitaxial CsPbBr₃ thin film.

Fig. R2 STEM images for different domains of CsPbBr₃/Gr/NaCl with zone axes of [001] (a) and [110] (b) and FFTs in their insets.

We agree that it is not necessary to repeat “for the first time” several times in the paper. We have removed the repeated ones in the main text and leave the ones in the abstract.

Associated changes to the main text and supplementary information (SI):

Figures: Fig. 4j and k (ferroelastic domains) have been added in revised main text. Fig. S26e and f (PFM mappings) have been added in SI.

The part in main text “Controlling domain wavelength via remote epitaxy” has been moved to SI page 6.

Main text page 2:

The words “for the first time” have been deleted:

“In this report, by taking advantages of the concept of remote epitaxy and its ability in controlling film-substrate coupling and further regulating dislocation densities (misfit and then threading), we have successfully epitaxially grown halide perovskite crystals and films of controlled dislocations (derived from the density of etching rosettes) on two strong polar NaCl and CaF₂ substrates with monolayer graphene buffered.”

Main text page 10:

The words “For the first time” have been deleted:

“Epitaxial halide perovskite has been synthesized via a novel remote epitaxy approach.”

The word “first” has been deleted:

“The controlling of dislocation (threading) density, which is estimated from etch rosettes, enables the unveiling of the dislocation-carrier dynamic relation in halide perovskite.”

SI page 6:

The words “for the first time” have been deleted:

“Here, in accordance with theoretical modeling, we have observed striped ferroelastic domains in CsPbBr₃ from both ionic and remote epitaxy (with different thicknesses), as shown in Fig. S26c and d.”

Reviewer #2 (Remarks to the Author):

Recommendation: Reject, but indicate to the authors that further work might justify a resubmission

The authors made halide perovskite thin films using the remote epitaxy approaches, investigated the several physical properties. The growth mechanism of halide perovskites was discussed in terms of the electrostatic interaction between the polar substrate and perovskite. With aid of the DFT calculations and a semi-quantitative modelling, flake-like formation of halide perovskites on graphene was explained. Later the authors argued that the striped ferroelastic domains appear wider in the remote epitaxy, explained by the continuum modelling. They concluded that the dislocations in halide perovskites are strong recombination centres based on the PL measurements.

Authors report some impressive findings, but I am worried that the conclusions were not so interesting and appealing to researchers of the halide perovskites. Also, some of their conclusions seem not strongly supported by the measurements. For instance, it is concluded that dislocations recombine the carriers strongly, based on the stronger PL strength of CsPbBr₃ on graphene. But it might be caused by the larger grain size, different thickness, or different surface orientations. Moreover, the thickness of the samples is not explicitly discussed in the manuscript. Therefore, I do not recommend this manuscript to be published in Nature Communications.

(1) Why two samples, CsPbBr₃/Gr/NaCl and CsPbBr₃/Gr/CaF₂ exhibit different growth patterns? Also, it is not crystal clear why CsPbBr₃/Graphen/CaFe₂ samples were not investigated further.

(3) $\Delta\mu$ seems not well defined in the manuscript.

(4) The derivation of equation 6 might not clear to the readers. Authors might explain the derivation in the supporting information.

(5) In figure 4, the dimension of the y-axis is missing.

(6) In figure S6, the dimension of the y-axis is missing.

Our response:

We thank the reviewer for carefully reading the manuscript and the helpful comments. The reviewer agreed some impressive findings in our report but worried about the impact of the conclusions to the halide perovskites' research community. We here explain how our conclusions/findings from this report would impact the halide perovskites' research community and others' one by one.

First conclusion/finding:

For the first time, the epitaxial halide perovskite thin film is achieved by a novel remote epitaxy approach.

Impact:

A success of high-quality thin film growth via epitaxy in most material family would impact the related research community or even human life, such as the growth of p-type GaN for blue light-emitting diode (LED)⁷ and high-temperature Fe-based superconductor⁸. The quality of thin film (defects such as dislocation) is directly related to device performance. Our group has already successfully achieved high-temperature ionic epitaxy of single-crystalline halide perovskite thin film⁹. High-quality epitaxial halide

perovskite thin film might be a promising candidate to further improve the photovoltaic performance of halide perovskite solar cells. In this work, we step forward to achieve remote epitaxial single-crystalline halide perovskite thin film, enabling the possibility of high-quality free standing halide perovskite thin film, as shown in Fig. R3 (Fig. 11 in revised main text). This would impact both halide perovskite photovoltaics and flexible electronics community.

Fig. R3 Photograph of the as-grown remote epitaxial thin film and exfoliated thin film.

Second conclusion/finding:

Unscreened polar substrate electrostatic potential in graphene coated NaCl(001) and CaF₂(001) substrates.

Impact:

DFT simulations yield modified electrostatic potentials of polar substrates coated with graphene and reduced interfacial interactions, which support the remote epitaxy in the present case. It would also stimulate the simulations research community to simulate different constructions using other materials as epilayer or using other 2D materials as buffer layer.

Third conclusion/finding:

Suppressing nucleation and promoting growth in remote epitaxy.

Impact:

By using a semi-quantitative modeling, we explain the experimental observations and understand the growth kinetics in remote epitaxy of halide perovskite. This fundamental understanding on the growth kinetics would guide the researchers to control the epitaxial growth of halide perovskite in this community.

Fourth conclusion/finding:

Dislocation-free mechanism in remote epitaxy (a new aspect added into the revised main text).

Impact:

By using molecular-dynamics (MD) computer simulations, we have revealed the growth mechanism and dislocation-formation mechanism in ionic epitaxy, remote epitaxy and van der Waals epitaxy. The formation of dislocation in ionic epitaxy is expected and it is a typical phenomenon in epitaxy growth to release the epitaxial strain. However, for both remote and van der Waal epitaxy, no dislocation can be formed from MD simulations. Instead, crystal islands would glide to each other in order to release the interfacial strain. Fig. R4a and b (Fig. S15a and b in revised SI) show side views of ionic epitaxy and remote epitaxy, revealing formation of misfit dislocations in ionic epitaxy and crystal islands gliding in remote epitaxy. Fig. R5 (Fig. S16 in revised SI) is the top view of ionic epitaxy, showing formation of threading dislocations. Fig. R6 and Fig. R7 (Fig. S17 and S18 in revised SI) are the top views of remote and van der Waals epitaxy and the gliding crystal islands are indicated with arrows. Videos 4, 5 and 6 show the side views of the entire growth processes of ionic epitaxy, remote epitaxy and van der Waals epitaxy, respectively. Videos 7, 8 and 9 are the top views of ionic epitaxy, remote epitaxy and van der Waals epitaxy, respectively. Details are discussed in revised main text page 8. This finding unveils the mystery of remote and van der Waals epitaxy and deepens fundamental understanding on the epitaxy. Dislocation free is a pie in the sky in epitaxy community.

Fig. R4 Side views of MD simulations for ionic epitaxy (a) and remote epitaxy (b).

Fig. R5 Top view of MD simulations for ionic epitaxy.

Fig. R6 Top view of MD simulations for remote epitaxy.

Fig. R7 Top view of MD simulations for van der Waals epitaxy.

Fifth conclusion/finding:

Misfit/threading dislocation-carrier dynamics relation.

Impact:

As reviewer #1 mentioned that the impact of dislocation density on the carrier lifetime is not a surprise, but it was justified to be shown in the defect-tolerance halide perovskite, it is an unexpected finding in the halide perovskite community. The effects of grain boundary¹⁰⁻¹⁵, interfaces¹⁶⁻¹⁸, points defects¹⁹⁻²² and phase impurity²³⁻²⁵ in halide perovskite on the carrier dynamics and device performances have been widely recognized and studied^{21, 26-29}. It is the time to unveil the dislocation-carrier dynamic relation in halide perovskite. We use a novel remote epitaxy approach to control the dislocation density, thereby study the dislocation-carrier dynamic relation. Remote epitaxy yields much less dislocation density, greatly improving the PL intensity and carrier lifetime. We reveal that dislocation is another critical defect on influencing halide perovskite's optoelectronic properties in addition to grain boundary, point defects and phase impurity. Our study calls for further investigations on their control to tune halide perovskite device performances.

Associated changes to the main text and supplementary information (SI):

A new aspect “Dislocation formation in ionic and remote epitaxy” has been added in revised main text.

Figures: Fig. 4 in revised main text and Fig. S15-18 in revised SI have been added.

Main text page 7:

“Dislocation formation in ionic and remote epitaxy

Dislocations are often harmful to device performance. Under thermodynamically favorable condition, dislocation-free crystals are accessible when the growth of crystals from solutions is at low supersaturation or from melts is weakly supercooled³⁰. However, dislocation formation is thermodynamically favored in highly mismatched heteroepitaxy system if the film thickness is above its critical thickness (in order to relax the misfit strain energy in epilayer³¹). Traditionally, a buffer layer can be introduced into epitaxial growth to reduce dislocation density³². By using a special buffer layer of graphene to reduce the interfacial energy between epilayer and substrate, the epitaxial growth mechanism (thermodynamics and kinetics) in remote epitaxy is significantly different from that in traditional epitaxy. Thus, different strain relaxation mechanism is expected. Molecular-dynamics (MD) simulations are utilized to study the growth and strain relaxation mechanisms in ionic epitaxy, remote epitaxy and van der Waals epitaxy. Fig. 5a, d and g shows MD simulations of the side views of ionic epitaxy, remote epitaxy and van der Waals epitaxy, respectively. The dislocations are indicated by blue arrows, as shown in Fig. 5a-c. Fig. S15a and Fig. S16 show the side and top views of the formation of dislocation in ionic epitaxy, respectively. Surprisingly, no dislocation can be formed in both remote and van der Waals epitaxy. In order to release strain energy between epilayer and substrate during the growth, large interfacial energy in ionic epitaxy enables formation of dislocations. Misfit dislocations at the interface are observed in the side view in Fig. S15a and threading dislocations are observed in top views in Fig. S16. For both remote and van der Waals epitaxy, small interfacial energy enables small crystal islands gliding to big ones. The gliding of crystal islands is indicated by arrows in the side view of Fig. S15b (remote epitaxy), top views of Fig. S17 (remote epitaxy) and Fig. S18 (van der Waals epitaxy). Top views of Fig. S16-18 also show the decrease of nucleation sites and the ability of gliding with decreasing interfacial energy (from ionic epitaxy, remote epitaxy to van der Waals epitaxy). Videos 4, 5 and 6 show the side views of the entire growth processes of ionic epitaxy, remote epitaxy and van der Waals epitaxy, respectively. Videos 7, 8 and 9 are the top views of ionic epitaxy, remote epitaxy and van der Waals epitaxy, respectively.”

Further comments from Reviewer #2:

Also, some of their conclusions seem not strongly supported by the measurements. For instance, it is concluded that dislocations recombine the carriers strongly, based on the stronger PL strength of CsPbBr₃ on graphene. But it might be caused by the larger grain size, different thickness, or different surface orientations. Moreover, the thickness of the samples is not explicitly discussed in the manuscript. Therefore, I do not recommend this manuscript to be published in Nature Communications.

Our response:

We thank reviewer’s comments. In the revised manuscript, we have added the thickness information of ionic epitaxial thin film (Fig. R8a (Fig. S5a in revised SI)), remote epitaxial thin films (Fig. R8b and c (Fig. S5b and c in revised SI)), ionic epitaxial flakes (Fig. R9a (Fig. S6a in revised SI)) and remote epitaxial flake (Fig. R9b (Fig. S6b in revised SI)). Here, the thin films for both remote (Fig. R8b) and ionic (Fig. R8a) epitaxy in our PL and TRPL measurements are approximately 1 μm. The flakes for both

remote (Fig. R9b) and ionic epitaxy (Fig. R9a) are also close. The grain size is around 1-5 μm , as estimated in Fig. 3 in main text and mentioned in revised main text page 9.

Fig. R8 Cross-sectional SEM images of CsPbBr₃/Gr/NaCl (a and b) and CsPbBr₃/NaCl (c) thin films.

Fig. R9 AFM images of flakes in CsPbBr₃/NaCl (a) and CsPbBr₃/Gr/NaCl (b), and thin films in CsPbBr₃/NaCl (c) and CsPbBr₃/Gr/NaCl (d).

We thank reviewer's comments on the possible reasons for the stronger PL strength and longer carrier lifetime of CsPbBr₃ on graphene, which might be larger grain size, different thickness, or different surface orientations. In the present work, the thickness of films and flakes for both remote and ionic epitaxial samples are similar, as shown in Fig. R8 and R9. The surface orientations are same based on the XRD reciprocal space mapping (RSM) in Fig. 1a and b in the revised main text for both ionic and remote epitaxial samples, respectively. The equation:

$$\frac{1}{\tau_{eff}} = \frac{1}{\tau_{bulk}} + \frac{1}{\tau_{int}} + \frac{1}{\tau_{gb}} + \frac{1}{\tau_{dis}} = \frac{1}{\tau_{bulk}} + \frac{1}{\frac{t}{2S_{int}} + \frac{1}{D}(\frac{t}{\pi})^2} + \frac{1}{\frac{d}{2S_{gb}} + \frac{1}{D}(\frac{d}{\pi})^2} + \frac{4\pi D\rho_D}{-\ln(\pi\rho_D r_0^2) - \frac{6}{5}} \quad (1)$$

has already taken the thickness and grain size into account. With the assumption that the recombination velocity at both interface and grain boundary are the same, we can study the dislocation recombination effect on the effective carrier lifetime, as shown in Fig. R10a and b (Fig. 5d in revised main text and Fig. S22b in revised SI) for different thickness and grain size, respectively. Since the grain size in our samples is estimated to be $\sim 1\text{-}5 \mu\text{m}$ (the grains for ionic epitaxy are smaller than the ones for remote epitaxy), it would not significantly change the carrier lifetime, as shown in Fig. R10b. Our data points are shown in Fig. R10a and b as well (thin films are in circle and flakes are in square), which fit the calculation well. Hence, we receive conclusion that dislocation strongly facilitates the recombination of the carriers and it

is as important as grain boundary or point defect in halide perovskite for physical properties and device performance.

Fig. R10 Effective carrier lifetime as a function of dislocation density at different sample thickness (a) and grain size (b). Experimental data for thin film and flake are indicated in circle and square, respectively. Remote epitaxy and ionic epitaxy regions are painted in dark purple and light purple, respectively.

Associated changes to the main text and supplementary information (SI):

Figures: Experimental data points have been added in Fig. 5d in revised main text. Fig. S5, S6 and S22 have been added in revised SI.

Main text page 4-5:

“Typical cross-sectional SEM images of the ionic epitaxial film with thickness of around 1 μm and the remote epitaxial films with thickness of around 1.5 μm , 1 μm and 2.2 μm , are shown in Fig. S5a, Fig. 1f, Fig. S5b and Fig. S5c, respectively. Atomic force microscopy (AFM) images of Fig. S6a and b show smooth surface of ionic and remote epitaxial flakes with surface root mean square (RMS) roughness of 0.9 nm and 0.4 nm at 3 μm lateral scale, respectively. The thickness of ionic and remote epitaxial flakes are estimated to be around 750 nm and 860 nm from the height profiles of Fig. S6e and f, respectively.”

Further comments from Reviewer #2:

(1) Why two samples, CsPbBr₃/Gr/NaCl and CsPbBr₃/Gr/CaF₂ exhibit different growth patterns? Also, it is not crystal clear why CsPbBr₃/Graphen/CaFe₂ samples were not investigated further.

Our response:

The exhibition for the different growth patterns in CsPbBr₃/Gr/NaCl and CsPbBr₃/Gr/CaF₂ is mainly due to different lattice misfits between different orientations of CsPbBr₃ and substrates. For

CsPbBr₃/Gr/NaCl(001) or CsPbBr₃/NaCl(001), along CsPbBr₃[100] || NaCl[100] direction, it has about 5% mismatch, which is much lower than other orientations. For CsPbBr₃/Gr/CaF₂ or CsPbBr₃/CaF₂, the mismatch along CsPbBr₃[011]||CaF₂[100] is 3.7%, which is remarkably smaller than the one along CsPbBr₃[100]||CaF₂[100] (9%). As a result, CsPbBr₃ grows with its out of plane as (001) orientation on NaCl or Gr/NaCl, while with its out of plane as (011) orientation on CaF₂ or Gr/CaF₂, combining with their surface free energy, resulting totally different growth patterns, as shown in Fig. R11a and b (Fig. 1e and g in main text), for CsPbBr₃/Gr/NaCl and CsPbBr₃/Gr/CaF₂, respectively.

Due to the reclined triangular prism grain in CsPbBr₃(011)/Gr/CaF₂ sample, as shown in Fig. R11b, it seems difficult to form smooth thin film after continuous growth. From the application point of view, it is not as interesting as CsPbBr₃(001)/Gr/NaCl for further investigation. However, the fact that remote epitaxy of halide perovskite can be achieved on different substrates (NaCl and CaF₂) implies that it is a universal approach and is possible to be applied in a versatile material family. The initial study of remote epitaxy of CsPbBr₃/Gr/CaF₂ enriches our understanding and shows reliability of our material growth. Nevertheless, we have done DFT calculations as well and added description/discussion in the revised main text page 6 and SI page 3 together with corresponding figures Fig. 2f-h in main text and Fig. S14c in SI.

Fig. R11 **a**, SEM image of remote epitaxial CsPbBr₃ flakes on Gr/NaCl. **b**, SEM image of remote epitaxial CsPbBr₃ triangular prisms on Gr/CaF₂.

Associated changes to the main text and supplementary information (SI):

Figures: Fig. 2f-h in revised main text, Fig. S14c and Fig. S22 in revised SI have been added.

Main text page 6:

“For CsPbBr₃(011) growth on Gr/CaF₂(001), the relaxed three layers of CsPbBr₃(011) lattice on monolayer graphene coated CaF₂(001) was chosen, as shown in Fig. S12c. The interlayer distances between CsPbBr₃(011) and graphene, and between graphene and CaF₂(001) are 2.75 Å and 2.26 Å respectively. Fig. 2f presents the corresponding charge transfer distribution between CsPbBr₃(011) and CaF₂(001) with monolayer graphene intercalation. From Fig. 2f we can observe that the CaF₂(001) substrate influences the charge distribution of graphene and subsequently modulates the growth of CsPbBr₃(011). Then the interfacial interaction energy between CsPbBr₃(011) and monolayer graphene coated CaF₂(001) was calculated as -27.83 meV/Å², indicating the graphene intercalation could effectively reduce the interfacial strain. The atomic stacking between top layer CaF₂(001) and graphene is presented in Fig. 2g. The electrostatic potential distribution contributed by the Ca atoms on the surface directly above graphene, with a distance of 2.7 Å is shown in Fig. 2h. In Fig. 2h, the observed pattern of

blue spots is consistent with $\text{CaF}_2(001)$ atomic stacking in Fig. 2g, implying the influence of $\text{CaF}_2(001)$ on the orientation of growth $\text{CsPbBr}_3(011)$.”

Main text page 10:

“By using equation (5) with constant values of sample thickness and grain size, the effective lifetime of halide perovskite CsPbBr_3 are plotted as a function of dislocation density, as shown in Fig. 5d and Fig. S22 for different sample thickness and grain size, respectively. Thus, the effect of dislocation density on the effective lifetime of halide perovskite CsPbBr_3 is quantitatively analyzed. The thickness and the grain size can influence the effective lifetime as well, as shown in lines with different colors in Fig. 5d and Fig. S22, respectively.”

SI page 3:

“For $\text{CsPbBr}_3(011)$ growth on $\text{Gr}/\text{CaF}_2(001)$, the calculation supercell was constructed as three layers of $\text{CsPbBr}_3(011)$ ($a=5.89 \text{ \AA}$, $b= 8.33 \text{ \AA}$) sitting on three layers of $\text{CaF}_2(001)$ ($a=5.35 \text{ \AA}$). The remote atomic interaction of $\text{CaF}_2(001)$ through monolayer graphene was simulated with $7a \times 2b$ $\text{CsPbBr}_3(011)$ sitting on monolayer graphene coated 8×3 $\text{CaF}_2(001)$. The selection of this model is mainly based on the minimization of the mismatch between CsPbBr_3 and $\text{CaF}_2(001)$. For the epitaxial relation of $\text{CsPbBr}_3(011) \parallel \text{CaF}_2(001)$, the mismatch is 3.8% along $\text{CsPbBr}_3[100] \parallel \text{CaF}_2[100]$ and -3.7% along $\text{CsPbBr}_3[0 \bar{1} 1] \parallel \text{CaF}_2[010]$, which is remarkably smaller than the mismatch of -9% along $\text{CsPbBr}_3[100] \parallel \text{CaF}_2[100]$ for $\text{CsPbBr}_3(001) \parallel \text{CaF}_2(001)$. The strain imposed on graphene is less than 2%. The top $\text{CaF}_2(001)$ layer and atoms above were allowed to relax until the forces on all the relaxed atoms were less than 0.05 eV/\AA . The vacuum gap in simulation supercell is larger than 20 \AA .”

(3) $\Delta\mu$ seems not well defined in the manuscript.

Our response:

Thanks for the comments. $\Delta\mu$ is the difference in volumetric Gibbs free energy of the two phases (solid and gas) (unit: meV/\AA^3). Because it is the driving force for the nucleation and growth, a general description of supersaturation for $\Delta\mu$ can be used.

Associated changes to the main text and supplementary information (SI):

Main text page 7:

“ $\Delta\mu$ is the difference in volumetric Gibbs free energy of the two phases (solid and gas) (unit: meV/\AA^3) and a general description of supersaturation for $\Delta\mu$ is used,”

(4) The derivation of equation 6 might not clear to the readers. Authors might explain the derivation in the supporting information.

Our response:

Thanks for the comments. The derivation of equation:

$$\Delta E_\alpha = -\frac{a_f(1+\alpha)}{a_s} \frac{\gamma_{int}}{t} + \epsilon \approx -\frac{a_f(1+\alpha)}{a_s} \frac{\gamma_{int}}{t} + \frac{1}{2} Y \alpha^2, (2)$$

is explained more details in SI. Epitaxial growth changes the total energy density of the simplified system ΔE_α , which is increased by the elastic strain energy density and released by bonding energy density. Assuming dangling bonds would be formed if a strain less than lattice misfit is applied, the ratio of bonding energy density to interfacial free energy density can be estimated to be the bonding atoms' percentage on the substrate, which is $a_f(1 + \alpha)/a_s$.

Associated changes to the main text and supplementary information (SI):

SI page 9:

“Assuming dangling bonds would be formed if a strain less than lattice misfit is applied, the ratio of bonding energy density to interfacial free energy density can be estimated to be the bonding atoms' percentage on the substrate, which is $a_f(1 + \alpha)/a_s$.”

(5) In figure 4, the dimension of the y-axis is missing.

Our response:

Thanks for the comments. The dimension of the y-axis has been added in Fig. S25a in revised SI (original Fig. 4).

(6) In figure S6, the dimension of the y-axis is missing.

Our response:

Thanks for the comments. The dimension of the y-axis has been added in Fig. S25b in revised SI (original Fig. S6).

Associated changes to the main text and supplementary information (SI):

Figure: The dimensions of the y-axis of Fig. S25a and b have been added in revised SI.

Reviewer #3 (Remarks to the Author):

This manuscript studies impact of remote epitaxy on the performance of halide perovskite. Since the discovery of remote epitaxy, it is the first work showing the improvement of performance in perovskite by performing remote epitaxy. The study must be meaningful for epitaxy community as well as halide-perovskite community. However, there are issues that authors must address before being considered for publication. Once all issues are completely address, the manuscript can be recommended for the publication as it contains very interesting physics and application potential.

Our response:

We thank the reviewer #3 for carefully reading the manuscript and the supportive comments. We appreciate the reviewer's comment that our study is meaningful for both epitaxy and halide-perovskite community. During last three months, we have tried our best to address the issues mentioned by the reviewers. We will response reviewer's comments one by one.

1. In the manuscript, main argument is about dislocation. However, there is no clear evidence showing dislocations. Instead it mainly shows electrical data. Authors should change their title. I suggest "Carrier lifetime enhancement in halide perovskite via remote epitaxy" to be considered. Title along this line fits better also because the biggest advantage from remote epitaxy in halide perovskite is an enhancement of carrier lifetime, which is very meaningful for halide perovskite community.

Our response:

We appreciate the reviewer's suggestion on the title "Carrier lifetime enhancement in halide perovskite via remote epitaxy" and we are certainly considering it in our revised manuscript. In the revised manuscript, we have done many HRTEM, HRAFM and low-voltage Cs-corrected STEM studies to reveal the dislocation information. We would like to hear back the reviewer's updated comment on the title after the reviewer reviews our revised manuscript. In our original manuscript, we have shown the evidence of dislocations from etching method. The etching method shows relatively low dislocation density, which is consistent with our calculation. Indeed, in the revised manuscript, in most cases, we receive perfect crystallite lattices, as shown in Fig. R12a-b (Fig. S10a-b in revised SI) obtained by STEM for CsPbBr₃/Gr/NaCl. Fig. R12c and d (Fig. S10c and d in revised SI) are inverse FFTs of their insets with highlighted white spots to highlight lattice fringes from Fig. R12b, indicating no dislocation. Nevertheless, after carefully searching from many images, we have observed dislocations in CsPbBr₃ flakes as shown in highlighted lattice fringes in Fig. R13d-f (Fig. S20d-f in revised SI). Fig. R13d-f are inverse FFTs of insets in Fig. R13a-c (Fig. S20a-c in revised SI). In addition, atomic-scale images of surface regions of remote epitaxial film obtained from HRAFM show both perfect crystallite lattices in Fig. R14a and b (Fig. S19a and b in revised SI) and threading dislocations in Fig. R14c and d (Fig. S19c and d in revised SI). We also got similar perfect crystallite lattices and threading dislocations in surface of ionic epitaxial film, as shown in Fig. R15a and b (Fig. S21a and b) and Fig. R15c and d (Fig. S21c and d), respectively.

Although we can find dislocations in these atomic-scale images, the density of dislocation cannot be estimated. Traditionally in semiconductors or other materials like NaCl³³, the threading dislocation density can be estimated by dislocation etch pits. We developed the etching technique for CsPbBr₃, which is similar to that for NaCl but with different concentration of etching solution. We observed etch rosettes to estimate the dislocation density. Based on these evidences on the dislocations, if the reviewer still

believes that the title “Carrier lifetime enhancement in halide perovskite via remote epitaxy” is more suitable for our study, it would also be our pleasure to take reviewer’s suggestion.

Fig. R12 STEM images for CsPbBr₃/Gr/NaCl at different regions with FFT in insets (a and b) and inverse FFTs (c and e) of their insets.

Fig. R13 HRTEM images for CsPbBr₃/Gr/NaCl at different regions (a-c) and inverse FFTs (d-f) of their insets. The insets in d-f are FFTs of a-c with additional white spots, respectively. Dislocations are indicated in d-f.

Fig. R14 HRAFM images for CsPbBr₃/Gr/NaCl at different surface regions (a and c) and inverse FFTs (b and d) of their insets. The insets in b and d are FFTs of a and c with additional white spots. Dislocations are indicated in d.

Fig. R15 HRAFM images for CsPbBr₃/NaCl at different surface regions (a and c) and inverse FFTs (b and d) of their insets. The insets in b and d are FFTs of a and c with additional white spots, respectively. Dislocations are indicated in d.

Associated changes to the main text and supplementary information (SI):

Figures: Fig. S10, S19-21 in revised SI have been added.

Main text page 8-9:

“STEM, HRAFM and HRTEM were utilized to characterize an as-grown remote epitaxial sample CsPbBr₃/Gr/NaCl. Perfect crystallite atomic structures of CsPbBr₃ are shown in cross-sectional images of Fig. 1k, Fig. S10a and b, and Fig. 4j and k obtained by Cs-corrected STEM. FFTs in their insets confirm the orthorhombic phase of CsPbBr₃. Two ferroelastic domains with zone axes of [001] and [1 $\bar{1}$ 0], as shown in Fig. 4j and k, respectively, are consistent with two out-of-plane peaks in the XRD ω -2 θ scan in Fig. S3. Fig. S10c and d are inverse FFTs of their insets with highlighted white spots to highlight lattice fringes from Fig. S10b, indicating no misfit dislocation. The atomic-scale image of a surface region of remote epitaxial film obtained from HRAFM exhibits perfect crystallite structure as well, as shown in Fig. S19a. Fig. 4i and Fig. S19b are the inverse FFTs of their insets to highlight atomic structure fringes from Fig. S10b and Fig. S19a, indicating no threading dislocation. However, after carefully searching of many images, threading dislocations can be observed in remote epitaxial CsPbBr₃ from some HRAFM and HRTEM images, as shown in highlighted lattice fringes in Fig. 19d, Fig. 4m and Fig. S20d-f (inverse

FFTs of their insets transformed from Fig. 19c, Fig. S20c and Fig. S20a-c, respectively). These threading dislocations observed from the remote epitaxial sample might be formed during large grain coalescence stage. More threading dislocations are found at a surface region of ionic epitaxial film, as shown in Fig. S21d (inverse FFT of its inset transformed from Fig. 21c), while similar perfect crystallite lattice is found at another surface region, as shown in Fig. S21b (inverse FFT of its inset transformed from Fig. 21a). It is noted that the density of dislocations could not be estimated from these atomic-scale images due to low density and uneven distribution of dislocations.”

2. There is no clear explanation regarding figure 5d. Authors need to clarify how the curve was obtained. In the manuscript, there are only two different dislocation density cases.

Our response:

The curve in Fig. R10a and b (Fig. 5d in main text) was obtained by using equation (1) (equation (5) in main text) with constant values of sample thickness and grain size. We assume the recombination velocity at both interface and grain boundary are the same (1.5×10^4 cm/s was taken³⁴). The diffusivity D is taken to be 0.35 cm²/s³⁴, τ_0 is taken to be 5.87×10^{-8} cm (the lattice constant of cubic CsPbBr₃). Detail explanation regarding Fig. 5d has been added in the main text in page 10. Our data points are indicated in Fig. R9a and b (Fig. 5d in main text). Although we only have two different dislocation density cases, they fit the calculations relatively well. In principle, by engineering the interfacial energy via either changing the substrate or the 2D buffer layer, one could obtain epitaxial thin film with different dislocation density thereby more data points would be obtained. Nevertheless, the two cases in our study are sufficient to show the trend and support our hypothesis on the dislocation-carrier dynamics relation.

Fig. R10 Effective carrier lifetime as a function of dislocation density at different sample thickness (a) and grain size (b). Experimental data for thin film and flake are indicated in circle and square, respectively. Remote epitaxy and ionic epitaxy regions are painted in dark purple and light purple, respectively.

Associated changes to the main text and supplementary information (SI):

Figures: Experimental data points have been added in Fig. 5b in revised main text and Fig. S22 in SI.

Main text page 10:

“By using equation (5) with constant values of sample thickness and grain size, the effective lifetime of halide perovskite CsPbBr_3 is plotted as a function of dislocation density, as shown in Fig. 5d and Fig. S22 for different sample thickness and grain size, respectively. Thus, the effect of dislocation density on the effective lifetime of halide perovskite CsPbBr_3 is quantitatively analyzed. The thickness and the grain size can influence the effective lifetime as well, as shown in lines with different colors in Fig. 5d and Fig. S22, respectively.”

3. Authors need to measure AFM on the surface of both ionic-epitaxially grown one and remote-epitaxially grown one to show how good morphology they have.

Our response:

Following the reviewer’s suggestion, we have performed AFM measurements on the surface of both thin films and flakes grown by ionic and remote epitaxy, as shown in Fig. R9a-d (Fig. S6a-d in SI), respectively. Both ionic epitaxial and remote epitaxial flakes show smooth surface with surface root mean square (RMS) roughness of around 0.9 nm and 0.4 nm, respectively, while rough surface could be found in ionic epitaxial thin film with surface RMS roughness of around 4.6 nm and improved surface in remote epitaxial thin film with RMS roughness of around 1.5 nm.

Fig. R9 AFM images of flakes in $\text{CsPbBr}_3/\text{NaCl}$ (a) and $\text{CsPbBr}_3/\text{Gr}/\text{NaCl}$ (b), and thin films in $\text{CsPbBr}_3/\text{NaCl}$ (c) and $\text{CsPbBr}_3/\text{Gr}/\text{NaCl}$ (d).

Associated changes to the main text and supplementary information (SI):

Main text page 4 and 5:

“Atomic force microscopy (AFM) images of Fig. S6a and b show smooth surface of ionic and remote epitaxial flakes with surface root mean square (RMS) roughness of 0.9 nm and 0.4 nm at 3 μm lateral scale, respectively. The thickness of ionic and remote epitaxial flakes are estimated to be around 750 nm and 860 nm from the height profiles of Fig. S6e and f, respectively. The surface morphology and height

profiles of both ionic and remote epitaxial films are shown in Fig. S6c, g and Fig. S6d, h, respectively. The surface RMS roughness for the ionic epitaxial film in Fig. S6c is calculated to be around 4.6 nm at 10 μm lateral scale, while it decreases to 1.5 nm for the remote epitaxial film in Fig. S6d.”

4. Since graphene’s properties can critical for remote epitaxy, authors should clearly reveal the properties of graphene such as crystallinity and uniformity. For example, laser confocal microscopy could be useful to confirm the uniformity of 2D materials.

Our response:

We strongly agree to reviewer’s comment that graphene’s properties are critical for remote epitaxy. Following reviewer’s advice, we have characterized the crystallinity and uniformity of graphene before epitaxial growth by using confocal Raman microscopy, as shown in Fig. R16 (Fig. S1 in SI). By using Raman mapping of I_{2D}/I_G and I_D/I_G of graphene, we observed both defected (dark area in Fig. R16c and g, bright area in Fig. R16d and h) and good (bright area in Fig. R16c and g, dark area in Fig. R16d and h) regions of graphene on both NaCl(001) and CaF₂(001).

Fig. R16 Optical images of transferred graphene on NaCl (a) and CaF₂ (e). Raman spectra of both defected and good regions of graphene on NaCl (b) and CaF₂ (f). Raman mapping of I_{2D}/I_G and I_D/I_G of graphene on NaCl(001) (c and d) and CaF₂(001) (g and h), respectively. The mapping area of 15 \times 15 μm^2 is indicated in green square in a and e.

We have further used HRAFM to characterize the atomic-scale structures of graphene on both NaCl and CaF₂ substrates, as shown in Fig. R17a and c (Fig. S2a and c in revised SI), respectively. Fig. R17b and d (Fig. S2b and d in revised SI) show FFTs and the bright spots can be indexed to graphene structure. We observed the spots splitting in Fig. R17b and the unclear additional spots in Fig. 17d, which might be induced by the chemical potential influence from the substrates underneath.

Fig. R17 HRAFM of graphene on NaCl(001) (a) and CaF₂(001) (c) and their FFTs (b and d, respectively).

Associated changes to the main text and supplementary information (SI):

Figures: Fig. S1 and S2 have been added in revised SI.

Main text page 3 and 4:

“Before growth, graphene layers are transferred onto NaCl(001) and CaF₂(001) as new substrates (Gr/NaCl(001) and Gr/NaCl(001)) for remote epitaxy. Optical images in supplementary information Fig. S1a and e show the surface morphology of Gr/NaCl(001) and Gr/NaCl(001). Both defected region³⁵ (low intensity ratio of 2D and G bands in Fig. S1c and g, and high intensity of D and G bands in Fig. S1d and h) and good region (high intensity ratio of 2D and G bands in Fig. S1c and g, and low intensity ratio of D and G bands in Fig. S1d and h) of graphene on substrates are characterized by Raman spectra and mappings, as shown in Fig. S1b-d and S1f-h. The atomic-scale structures of graphene surface on both NaCl(001) and CaF₂(001) substrates are investigated by high resolution atomic force microscopy (HRAFM), as shown in Fig. S2a and c, respectively. Fig. S2b and d show the fast Fourier transforms (FFT) and the bright spots can be indexed to graphene structure. The spots splitting in Fig. S2b and the unclear additional spots in Fig. S2d may be induced by the chemical potential influence from the substrates underneath.”

5. After exfoliating epilayers from graphene, authors must show the quality of graphene, morphology of substrate, exfoliation yield, and condition of epilayers. This confirmation is important to make an argument of remote epitaxy on graphene. Thorough investigation using AFM, SEM, and TEM is needed.

Our response:

We demonstrated the possibility of exfoliating epilayer from graphene, as shown in Fig. R3 (Fig. 1i in revised main text). From the optical microscopy images, as shown in Fig. R18a and b, we can see the exfoliation yield is high, but the film is cracked (Fig. R18b). One possible reason is that the halide perovskite film is brittle when it is thick. Another possible reason is that the defects on graphene would hinder the exfoliation due to the strong bonding between the film and defects, as shown in remained crystals on the substrate after exfoliation in Fig. R18a. The graphene may remain on the substrate or film. We characterized graphene remained on the substrate after exfoliation by using Raman mapping, as shown in Fig. R18c-e. There is no clear change of quality in graphene after growth and exfoliation except that there are holes due to the exfoliation process. The quality of epilayer after exfoliation (transferred onto TEM grid) at local regions can be confirmed in the HRTEM images, as shown in Fig. R13 (Fig. S20 in revised SI). Hence, from the application point of view, there are still many issues to be addressed in order to achieve flexible electronic device based on remote epitaxial halide perovskite. However, at least, the remote epitaxy exhibits the possibility to obtain free standing films and flakes and it is a promising approach to be applied in flexible electronic device fabrication. Once these issues are addressed, such as uniformity of graphene or other 2D materials, the remote epitaxy approach might have big impact on the industry in the near future.

Fig. R3 Photograph of the as-grown remote epitaxial thin film and exfoliated thin film.

Fig. R18 Optical images of remote epitaxial CsPbBr₃ on substrate (a) and tape (b) after exfoliation. Zoom-in optical image (c) of substrate after exfoliation. Raman mapping of I_{2D}/I_G (d) and I_D/I_G (e) of graphene after exfoliation. The mapping area of $15 \times 15 \mu\text{m}^2$ is indicated in green square in c.

Fig. R13 HRTEM images for CsPbBr₃/Gr/NaCl at different regions (a-c) and inverse FFTs (d-f) of their insets. The insets in d-f are FFTs of a-c with additional white spots, respectively. Dislocations are indicated in d-f.

6. Authors must discuss about the full detail of the process including method of graphene transfer on the substrate and condition of substrate after the transfer since they can critically impact on the quality of remote epitaxial layers. Characterization data in each process can be displayed in the supplementary. Also it would be great to show quality of epilayer depending on the transfer methods.

Our response:

The graphene used in this work was purchased from Graphene Laboratories Inc. (Calverton, New York, US). The graphene was synthesized on Cu foils by chemical vapor deposition. For transfer of graphene, poly(methyl methacrylate) (PMMA) was spin-coated and baked on graphene/Cu to protect the graphene on the front side of Cu foils. The rear side of Cu foils was then treated in O₂ plasma etcher to remove unwanted graphene. Afterwards, the PMMA/graphene/Cu stack was placed in ammonium persulfate aqueous solution (60 g/L) to etch Cu. Once Cu was etched, the PMMA/graphene was rinsed in water (isopropyl alcohol) several times and then scooped out using the CaF₂ (NaCl) substrate. The PMMA/graphene/CaF₂(NaCl) was dried in air. In the end, PMMA was dissolved in acetone.

The condition of substrate and graphene after the transfer has been discussed in our reply to last comment. The transfer process is well developed in our group and literature as well^{1, 36-38}. Our results (Raman mapping) in Fig. R16 is similar to that in Fig. R19 from the literature with limited good region of around

few micrometers. We are eager to develop new transfer method to achieve better quality of graphene thereby to enhance the quality of epilayer. It seems not very necessary to use other transfer methods that give worse quality of graphene to do epitaxy growth for comparison.

Fig. R19 Optical microscopy (a) and micro-Raman spectroscopy images (b-d)¹.

Associated changes to the main text and supplementary information (SI):

SI page 2:

“The graphene used in this work was purchased from Graphene Laboratories Inc. (Calverton, New York, US). The graphene was synthesized on Cu foils by chemical vapor deposition. For transfer of graphene, poly(methyl methacrylate) (PMMA) was spin-coated and baked on graphene/Cu to protect the graphene on the front side of Cu foils. The rear side of Cu foils was then treated in O₂ plasma etcher to remove unwanted graphene. Afterwards, the PMMA/graphene/Cu stack was placed in ammonium persulfate aqueous solution (60 g/L) to etch Cu. Once Cu was etched, the PMMA/graphene was rinsed in water (isopropyl alcohol) several times and then scooped out using the CaF₂ (NaCl) substrate. The PMMA/graphene/CaF₂(NaCl) was dried in air. In the end, PMMA was dissolved in acetone.”

7. What could happen if interface between the substrate and graphene is contaminated? It could be interesting if authors include the experimental result to show some interface condition.

Our response:

If interface between the substrate and graphene is contaminated, it is expected that the contamination would fully screen the chemical potential from the polar substrates. Hence, no epitaxy growth can be achieved at the contaminated regions. Since the transferred graphene is not perfect, there is indeed contaminated or defected area on substrates. The morphology of the contaminated or defected region is totally different from the epitaxial region after growth and can be easily distinguished by morphology (e.g. shape of flake) from optical microscopy images. Fig. R20 shows the morphology of a sample and epitaxial flakes are well arranged. Some typical contaminated or defected regions are indicated in blue circle, resulting in no growth or none epitaxial growth.

Fig. R20 Optical microscopy image of remote epitaxial CsPbBr₃ flakes on Gr/NaCl(001). Some contaminated or defected regions are indicated in blue circles.

Following the reviewer's suggestion, we have characterized the graphene at the interface by using Raman mapping, as shown in Fig. R21a-f (Fig. S7a-f in revised SI). The graphene G-band and 2D-band can be seen in Raman spectra in Fig. R21b after growth. Besides, additional D-band of graphene can be found under CsPbBr₃ flakes, which might be related to the disorder induced by epitaxial growth. Fig. R21f shows a slightly shift towards high wavenumber in 2D-band under flakes, indicating compression in graphene due to epitaxial strain. What kind of disorders or defects was induced into the graphene at the interface is still a puzzle for us.

Fig. R21 Optical image (a), Raman spectrum (b), Raman mapping (c) of remote epitaxial CsPbBr₃ flakes. The mapping area (10×10 μm²) is indicated in green square in a. Raman mapping of I_{2D}/I_G (d), I_D (e) and 2D peak position (f) of graphene.

Associated changes to the main text and supplementary information (SI):

Figures: Fig. S7 has been added in revised SI.

Main text page 5:

One paragraph about the Raman spectrum and mapping on the remote epitaxial sample has been added in the revised main text, which is

“The as-grown CsPbBr₃ flakes in remote epitaxy and the status of graphene after growth are characterized by Raman spectra and mapping, as shown in Fig. S7b and c-f, respectively. The Raman mapping region (10×10 μm²) is indicated in green square in Fig. S7a. The Raman peaks at around 68 cm⁻¹, 120 cm⁻¹ and 307 cm⁻¹ confirm the orthorhombic phase of CsPbBr₃³⁹, as shown in Fig. S7b and c. The graphene G-band and 2D-band can still be seen in Raman spectra in Fig. S7b after growth. Besides, additional D-band of graphene can be found under CsPbBr₃ flakes, which might be related to the disorder induced by epitaxial growth. Fig. S7f shows a slightly shift towards high wavenumber in 2D-band under flakes, indicating compression in graphene due to epitaxial strain.”

7. It is hard to know nucleation rate in case of ionic epitaxy through data in the manuscript. Authors need to include some SEM images of the ionic epitaxy case like figure 1e/g

Our response:

The SEM images of the ionic epitaxial thin films are shown in Fig. R22a and b (Fig. 3d and left side of Fig. 3e in main text). It is indeed hard to estimate the nucleation rate in the case of ionic epitaxy once thin film is formed. We could also find some regions that give ionic epitaxial flakes, as shown in Fig. R22c and d. Based on our semi-quantitative modeling, the nucleation rate in the case of ionic epitaxy is two orders magnitude higher than the case of remote epitaxy. So at the same region of the substrate (same superstation), thin film (coalesced flakes) is preferred to be formed at the region without graphene buffered (the case of ionic epitaxy) even only a few flakes are formed at the region with graphene buffered (the case of remote epitaxy).

Fig. R22 SEM images of ionic epitaxial CsPbBr₃ thin films (a and b). Optical images of ionic epitaxial CsPbBr₃ thin films (c and d).

8. Detail study in the CaF₂ case could be meaningful for obtaining general idea for remote heteroepitaxy. DFT calculation, TRPL study can strengthen significantly.

Our response:

We agree that detail study in the CaF₂ case could be meaningful for a general idea for remote heteroepitaxy. In the revised manuscript, we have conducted DFT calculation of CsPbBr₃(011) growth on Gr/CaF₂(001) and added it in the revised main text and SI, as shown in Fig. R23 (Fig. 2f-h and Fig. S14 in main text and SI, respectively). The calculation supercell was constructed as three layers of CsPbBr₃(011) ($a=5.89 \text{ \AA}$, $b=8.33 \text{ \AA}$) sitting on three layers of CaF₂(001) ($a=5.35 \text{ \AA}$). The remote atomic interaction of CaF₂(001) through monolayer graphene was simulated with $7a \times 2b$ CsPbBr₃(011) sitting on monolayer

graphene coated 8×3 $\text{CaF}_2(001)$. The selection of this model is mainly based on the minimization of the mismatch between CsPbBr_3 and $\text{CaF}_2(001)$. For the epitaxial relation of $\text{CsPbBr}_3(011) \parallel \text{CaF}_2(001)$, the mismatch is 3.8% along $\text{CsPbBr}_3[100] \parallel \text{CaF}_2[100]$ and -3.7% along $\text{CsPbBr}_3[0\bar{1}1] \parallel \text{CaF}_2[010]$, which is remarkably smaller than the mismatch of -9% along $\text{CsPbBr}_3[100] \parallel \text{CaF}_2[100]$ for $\text{CsPbBr}_3(001) \parallel \text{CaF}_2(001)$. The strain imposed on graphene is less than 2%. The top $\text{CaF}_2(001)$ layer and atoms above were allowed to relax until the forces on all the relaxed atoms were less than 0.05 eV/\AA . The vacuum gap in simulation supercell is larger than 20 \AA .

Fig. R23 Remote atomic interaction between $\text{CsPbBr}_3(011)$ and graphene coated $\text{CaF}_2(001)$. (a) DFT simulation of $\text{CsPbBr}_3(011)$ on monolayer graphene coated $\text{CaF}_2(001)$. (b) Charge transfer distribution between $\text{CsPbBr}_3(011)$ and $\text{CaF}_2(001)$ with graphene intercalation. (c) Atomic stacking between $\text{CaF}_2(001)$ top layer and coated monolayer graphene. (d) Potential fluctuation at the epitaxial surface from $\text{CaF}_2(001)$ through monolayer graphene, where blue pattern is consistent with center site of four Ca atoms in (c).

Figure R23a is the relaxed three layers of $\text{CsPbBr}_3(011)$ lattice on monolayer graphene coated $\text{CaF}_2(001)$. The interlayer distances between $\text{CsPbBr}_3(011)$ and graphene, and between graphene and $\text{CaF}_2(001)$ are 2.75 \AA and 2.26 \AA respectively. Fig. R23b presents the corresponding charge transfer distribution between $\text{CsPbBr}_3(011)$ and $\text{CaF}_2(001)$ with monolayer graphene intercalation. From Fig. R23b we can observe that the $\text{CaF}_2(001)$ substrate influences the charge distribution of graphene and subsequently modulates the growth of $\text{CsPbBr}_3(011)$. Then the interfacial interaction energy between $\text{CsPbBr}_3(011)$ and monolayer graphene coated $\text{CaF}_2(001)$ was calculated as -27.83 meV/\AA^2 , indicating the graphene intercalation could effectively reduce the interfacial strain. The atomic stacking between top layer $\text{CaF}_2(001)$ and graphene is presented in Fig. R23c. The electrostatic potential distribution contributed by the Ca atoms on the surface directly above graphene, with a distance of 2.7 \AA is shown in Fig. R23d. In Fig. R23d, the observed pattern of blue spots is consistent with $\text{CaF}_2(001)$ atomic stacking in Fig. R23c, implying the influence of $\text{CaF}_2(001)$ on the orientation of growth $\text{CsPbBr}_3(011)$.

For the TRPL study on both $\text{CsPbBr}_3(011)/\text{CaF}_2(001)$ and $\text{CsPbBr}_3(011)/\text{Gr}/\text{CaF}_2(001)$, we observed similar phenomenon as that in the case of $\text{CsPbBr}_3(001)/\text{Gr}/\text{NaCl}(001)$, i.e. TRPL shows enhancement of the effective carrier lifetime in remote epitaxial CsPbBr_3 , as shown in Fig. R24.

Fig. R24 TRPL of CsPbBr₃ on CaF₂ in both remote and ionic epitaxy.

Associated changes to the main text and supplementary information (SI):

Figures: Fig. 2f-h in revised main text, Fig. S14 and S23 in revised SI have been added.

Main text page 6:

“For CsPbBr₃(011) growth on Gr/CaF₂(001), the relaxed three layers of CsPbBr₃(011) lattice on monolayer graphene coated CaF₂(001) was chosen, as shown in Fig. S14c. The interlayer distances between CsPbBr₃(011) and graphene, and between graphene and CaF₂(001) are 2.75 Å and 2.26 Å respectively. Fig. 2f presents the corresponding charge transfer distribution between CsPbBr₃(011) and CaF₂(001) with monolayer graphene intercalation. From Fig. 2f we can observe that the CaF₂(001) substrate influences the charge distribution of graphene and subsequently modulates the growth of CsPbBr₃(011). Then the interfacial interaction energy between CsPbBr₃(011) and monolayer graphene coated CaF₂(001) was calculated as -27.83 meV/Å², indicating the graphene intercalation could effectively reduce the interfacial strain. The atomic stacking between top layer CaF₂(001) and graphene is presented in Fig. 2g. The electrostatic potential distribution contributed by the Ca atoms on the surface directly above graphene, with a distance of 2.7 Å is shown in Fig. 2h. In Fig. 2h, the observed pattern of blue spots is consistent with CaF₂(001) atomic stacking in Fig. 2g, implying the influence of CaF₂(001) on the orientation of growth CsPbBr₃(011).”

Main text page 10:

“Similar enhancement of the effective lifetime is also observed in the case of CsPbBr₃ grown on Gr/CaF₂, as shown in Fig. S23.”

SI page 3:

“For CsPbBr₃(011) growth on Gr/CaF₂(001), the calculation supercell was constructed as three layers of CsPbBr₃(011) (a=5.89 Å, b= 8.33 Å) sitting on three layers of CaF₂(001) (a=5.35 Å). The remote atomic interaction of CaF₂(001) through monolayer graphene was simulated with 7a×2b CsPbBr₃(011) sitting on monolayer graphene coated 8×3 CaF₂(001). The selection of this model is mainly based on the minimization of the mismatch between CsPbBr₃ and CaF₂(001). For the epitaxial relation of CsPbBr₃(011)∥CaF₂(001), the mismatch is 3.8% along CsPbBr₃[100]∥CaF₂[100] and -3.7% along CsPbBr₃[0 1 1]∥CaF₂[010], which is remarkably smaller than the mismatch of -9% along CsPbBr₃[100]∥CaF₂[100] for CsPbBr₃(001)∥CaF₂(001). The strain imposed on graphene is less than 2%. The top CaF₂(001) layer and atoms above were allowed to relax until the forces on all the relaxed atoms were less than 0.05 eV/Å. The vacuum gap in simulation supercell is larger than 20 Å.”

9. Epitaxial surface does not seem to be atomically flat. How about the homogeneity in terms of carrier transportation? Does the remote-epitaxially grown one have a better homogeneity? Conducting AFM mapping could be useful to figure out this issue.

Our response:

In general, the flake is atomically flat but the coalesced film is not atomically flat. The epitaxial surface of thin films for both ionic and remote epitaxy is not atomically flat, as shown in Fig. R9c and d (Fig. S6c and d in SI), respectively. The surface RMS roughness of ionic epitaxial thin film is around 4.6 nm, while that of remote epitaxial thin film is around 1.5 nm. Even the surface of single flake does not seem to be atomically flat, as shown in SEM image Fig. 3g and h in main text. However, the morphology shown in SEM might not be the original one due to the charging effect (insulator substrates NaCl and CaF₂) or degradation by the e-beam (instability of halide perovskite). From the AFM measurements shown in Fig. R9a and b (Fig. S6a and b) for ionic and remote epitaxial flake, respectively, we have obtained atomically flat surface with calculated surface RMS roughness of 0.9 nm for ionic one and 0.4 nm for remote one at 3 μm lateral scale. Also, we need to mention that, there are steps on the surface of single flake as well and between steps, it is atomically flat. Overall, both remote epitaxial thin film and flake show slightly improved surface roughness and better homogeneity. The atomically flat surface at local area has also been confirmed by HRAFM, as shown in Fig. R14a and c and Fig. R15 a and c, for ionic and remote epitaxial flake, respectively. In terms of carrier transportation, we could not get any results from conducting AFM measurements because the substrates of NaCl and CaF₂ are not conductive. We may seek for such study on transferred sample without surface damage in future.

Fig. R9 AFM images of flakes in CsPbBr₃/NaCl (a) and CsPbBr₃/Gr/NaCl (b), and thin films in CsPbBr₃/NaCl (c) and CsPbBr₃/Gr/NaCl (d).

Fig. R14 HRAFM images for CsPbBr₃/Gr/NaCl at different surface regions (a and c) and inverse FFTs (b and d) of their insets. The insets in b and d are FFTs of a and c with additional white spots. Dislocations are indicated in d.

Fig. R15 HRAFM images for CsPbBr₃/NaCl at different surface regions (a and c) and inverse FFTs (b and d) of their insets. The insets in b and d are FFTs of a and c with additional white spots, respectively. Dislocations are indicated in d.

Associated changes to the main text and supplementary information (SI):

Figures: Fig. S6, S19 and S21 have been added in revised SI.

Main text page 3-4:

“AFM images of Fig. S6a and b show smooth surface of ionic and remote epitaxial flakes with surface root mean square (RMS) roughness at 3 μm lateral scale of 0.9 nm and 0.4 nm, respectively. The thickness of ionic and remote epitaxial flakes are estimated to be around 750 nm and 860 nm from the height profiles of Fig. S6e and f, respectively. The surface morphology and its height profile of both ionic and remote epitaxial films are shown in Fig. S6c, g and Fig. S6d, h, respectively. The surface RMS roughness at 10 μm lateral scale for the ionic epitaxial film in Fig. S6c is calculated to be around 4.6 nm, while it decreases to 1.5 nm for the remote epitaxial film in Fig. S6d.”

10. Actually, dislocation is just one type of defects. Authors claim that all effect comes from dislocation reduction. If so, authors have to study more to show the effect from dislocation distinguished from other types of defects.

Our response:

Yes, dislocation is just one type of defects. The effect on PL strength and lifetime could also come from grain boundary¹⁰⁻¹⁵, interfaces¹⁶⁻¹⁸, points defects¹⁹⁻²² and phase impurity²³⁻²⁵. In our case of epitaxial halide perovskite grown by high-temperature vapor phase, the film has much higher crystal quality than the low-temperature solution-based one⁹. Hence, we can expect that the effects of point defects and phase impurity are limited and might not be a dominant contribution. We also take the grain boundary (grain size) and interface into account to calculate the effective lifetime, which is included in equation (1) (equation (5) in main text). Based on our calculation according to the equation (1), the grain boundary (grain size) and interface indeed play an important role, as shown in different curve in Fig. R10 (Fig. S22). However, in our case, grain size, if there is, and interfaces are tried to be kept constant during characterization. The PL enhancement and longer lifetime in remote epitaxial samples thus could be attributed to the dislocation reduction. Our experimental data fits our hypothesis/calculations on the effective lifetime well, as shown in Fig. R10. As a result, we tend to attribute major change in PL from dislocation reduction but we agree that small difference in grain boundary and interfaces could contribute slightly.

Fig. R10 Effective carrier lifetime as a function of dislocation density at different sample thickness (a) and grain size (b). Experimental data for thin film and flake are indicated in circle and square, respectively. Remote epitaxy and ionic epitaxy regions are painted in dark purple and light purple, respectively.

Associated changes to the main text and supplementary information (SI):

Figures: Experimental data points have been added in Fig. 5d in main text and Fig. S22 in revised SI.

11. TEM at the epitaxial interface is highly recommended

Our response:

We agree to the reviewer's advice. Indeed, in past three months, we have tried our best effort on preparing TEM samples and image the samples with HRTEM and Cs-corrected low voltage STEM. We did FIB for TEM samples at Tsinghua University and Brown University, as shown in Fig. R25 a-d for ionic and remote epitaxial samples, respectively. One typical sample is shown in Fig. R26a (Fig. S11a) from STEM. We did low-voltage (120 kV) HRTEM at Tsinghua University, high-voltage (200 kV) HRTEM at Brown University and Connell University. We did low-voltage (80 kV) STEM at Ernst Ruska-Centre for Microscopy and Spectroscopy with Electrons in Germany.

Fig. R25 SEM images of FIB samples for CsPbBr₃/NaCl (a and b) and CsPbBr₃/Gr/NaCl (c and d).

Fig. R26 STEM images of CsPbBr₃/Gr/NaCl (a) and NaCl (b and c).

From low-voltage STEM, we captured smooth interface between CsPbBr₃ (bright part), NaCl (dark part) and Gr (dark line in between) and atomic-resolution image of epilayer CsPbBr₃ in CsPbBr₃/Gr/NaCl, as shown in Fig. R27 (Fig. 1k in main text). The fast Fourier transform (FFT) in the inset confirms the

orthorhombic phase of CsPbBr₃. More atomic-resolution images of epilayer CsPbBr₃ and their FFTs are shown in Fig. R11 (Fig. S10a-d in SI) and their insets, respectively. We noted that NaCl in Fig. R26 is amorphous because the NaCl is extremely unstable even under low voltage of electron beam, as shown in Fig. R23a-c (Fig. S11a-c in SI) with holes formed. Comprehensive study of stability of both CsPbBr₃ and NaCl was carried out by in-situ HRTEM, as shown in Fig. R27 (Fig. S12 in SI) for CsPbBr₃/Gr/NaCl and Fig. R28 (Fig. S13 in SI) for NaCl in another region. The variation of spots in FFTs in the insets of Fig. R27 and Fig. R28 indicates both CsPbBr₃ and NaCl are unstable and CsPbBr₃ is better than NaCl. Video 1, 2 and 3 in supplementary information show the detail amorphization processes of bright field images of CsPbBr₃/Gr/NaCl and NaCl and diffraction pattern of NaCl, respectively.

Fig. R27 STEM image of CsPbBr₃/Gr/NaCl with FFT in inset.

Fig. R12 STEM images for CsPbBr₃/Gr/NaCl at different regions with FFT in insets (a and b) and inverse FFTs (c and e) of their insets.

Fig. R28 TEM images of CsPbBr₃/Gr/NaCl and FFTs in their insets with increasing time.

Fig. R29 TEM images of NaCl and FFTs in their insets with increasing time.

Associated changes to the main text and supplementary information (SI):

Figures: Fig. 1k and Fig. 10-12 have been added in revised main text and revised SI.

Videos: video 1, 2 and 3 have been added in revised SI.

Main text page 5:

“The as-grown epilayer can be easily exfoliated. The exfoliation process is shown in the schematic illustration in Fig. 1h and the corresponding photograph is shown in Fig. 1i. The exfoliated sample can be transferred on to transmission electron microscopy (TEM) grid. TEM of halide perovskite materials is still a challenge due to the degradation under electron beam⁴⁰. Before degradation, the TEM diffraction pattern is obtained, as shown in Fig. 1k. The diffraction spots can be well indexed to support the orthorhombic structure, and the zone axis is calculated to be [010]. The cross-sectional interfacial morphologies and crystallite structures were evaluated by low-voltage Cs-corrected scanning transmission electron microscope (STEM). The STEM samples of CsPbBr₃/Gr/NaCl were prepared by focused ion beam (FIB) and details are shown in the materials and methods part of supplementary information. Fig. 1j demonstrates smooth interfaces between CsPbBr₃ (bright part), NaCl (dark part) and Gr (dark line in between) and atomic-resolution image of epilayer CsPbBr₃ in CsPbBr₃/Gr/NaCl. The fast Fourier transform (FFT) in the inset confirms the orthorhombic phase of CsPbBr₃. More atomic-resolution images of epilayer CsPbBr₃ and their FFTs are shown in Fig. S10a-d and their insets, respectively. It is noted that NaCl in Fig. 1k is amorphous because the NaCl is extremely unstable even under low voltage of electron beam, as shown in Fig. S11a-c with holes formed. Comprehensive study of stability of both CsPbBr₃ and NaCl was carried out by in-situ HRTEM, as shown in Fig. S12 for CsPbBr₃/Gr/NaCl and Fig. S13 for

NaCl in another region. The variation of spots in FFTs in the insets of Fig. S12 and Fig. S13 indicates both CsPbBr₃ and NaCl are unstable and CsPbBr₃ is better than NaCl. Video 1, 2 and 3 in supplementary information show the detail amorphization processes of bright field images of CsPbBr₃/Gr/NaCl and NaCl and diffraction pattern of NaCl, respectively.”

12. Is there any usefulness when ferroelastic domain pattern changes? This can be very interesting.

Our response:

Yes, this is a very interesting finding. Since the fundamental physics of extraordinary optoelectronic performance of halide perovskite solar cells might be related to some domain physics^{2, 3}, it is very interesting and useful on studying their domain patterns. The domain engineering might be a promising direction for the enhancement of the performance of halide perovskite solar cells. Our finding on the controlling ferroelastic domain via remote epitaxy might be a big impact on the halide perovskite community.

Associated changes to the main text and supplementary information (SI):

As suggested by reviewer #1, the aspect “Controlling domain wavelength via remote epitaxy” has been moved to SI in page 9-10.

13. In addition to lifetime measurement, device demonstration can strengthen the paper significantly.

Our response:

Following the reviewer’s suggestion, we have made a photodetector with two gold stripe contacts deposited onto the CsPbBr₃ thin film with a physical mask on top by e-beam evaporation (Au/CsPbBr₃(remote epitaxial thin film)/Au), as shown in Fig. R30a (Fig. S24a in SI). The photo responses under UV light (405 nm laser, ~0.5 mW) are investigated for both devices made by ionic and remote epitaxial thin films, as shown in Fig. R30b (Fig. S24b in SI). Fig. R30b and d (Fig. S24c and d in SI) are the zoom-in rising and falling parts in Fig. R30b, respectively, revealing a faster rise time in remote epitaxial thin film based device (Fig. R30c) and an additional decay tail with a longer decay time of around 8.08 s in ionic epitaxial thin film based device (Fig. R30d). It also can be found that the dark current in the remote epitaxial thin film based device is two orders of magnitude lower than that in the ionic epitaxial thin film based device. These observations, fast rise time and disappearance of long decay tail in remote epitaxial thin film based device, could be attributed to the low density of dislocations in remote epitaxial thin film. Similar observations can be found in other photodetectors, such as AlGaN⁴¹.

Fig. R30 a, A photodetector (Au/CsPbBr₃/Au) made by gold contacts on top of thin film. b, Photo response on both devices made by ionic and remote epitaxial thin film. Enlarged rising (c) and falling (d) parts of current for both devices.

Associated changes to the main text and supplementary information (SI):

Figures: Fig. S24 has been added in revised SI.

Main text page 10-11:

“Two gold stripe contacts are deposited onto the CsPbBr₃ thin film with a physical mask on top by e-beam evaporation and a simple device of photodetector (Au/CsPbBr₃(remote epitaxial thin film)/Au) is fabricated, as shown in Fig. S24a. The photo responses under UV light (405 nm laser, ~0.5 mW) are investigated for both devices made by ionic and remote epitaxial thin films, as shown in Fig. S24b. Fig. S24c and d are the enlarged rising and falling parts in Fig. S24b, respectively, revealing a faster rise time in remote epitaxial thin film based device (Fig. S24c) and an additional decay tail with a longer decay time of around 8.08 s in ionic epitaxial thin film based device (Fig. S24d). Similar observations can be found in other photodetectors, such as AlGa^{N41}.”

14. STM could be powerful to observe the potential influence coming from NaCl or CaF₂.

Our response:

Thanks for the suggestion. Due to the insulator nature of substrates in our case, STM cannot be performed. Instead, we performed HRAFM on Gr/NaCl and Gr/CaF₂, as shown in Fig. R17 (Fig. S2 in revised SI).

Fig. R17 HRAFM of graphene on NaCl(001) (a) and CaF₂(001) (c) and their FFTs (b and d, respectively).

15. English correction is needed from a native speaker.

Our response:

A native speaker has corrected the English in the revised main text and SI.

Besides, all the changes are in red in the revised main text and SI.

Response References

1. Li X, Zhu Y, Cai W, Borysiak M, Han B, Chen D, *et al.* Transfer of Large-Area Graphene Films for High-Performance Transparent Conductive Electrodes. *Nano letters* 2009, **9**(12): 4359-4363.
2. Liu Y, Collins L, Proksch R, Kim S, Watson BR, Doughty B, *et al.* Chemical nature of ferroelastic twin domains in CH₃NH₃PbI₃ perovskite. *Nature materials* 2018, **17**(11): 1013-1019.
3. Huang B, Kong G, Esfahani EN, Chen S, Li Q, Yu J, *et al.* Ferroic domains regulate photocurrent in single-crystalline CH₃NH₃PbI₃ films self-grown on FTO/TiO₂ substrate. *npj Quantum Materials* 2018, **3**(1): 30.
4. Röhm H, Leonhard T, Hoffmann MJ, Colsmann A. Ferroelectric domains in methylammonium lead iodide perovskite thin-films. *Energy & Environmental Science* 2017, **10**(4): 950-955.
5. Strelcov E, Dong Q, Li T, Chae J, Shao Y, Deng Y, *et al.* CH₃NH₃PbI₃ perovskites: Ferroelasticity revealed. *Science Advances* 2017, **3**(4): e1602165.
6. Kutes Y, Ye L, Zhou Y, Pang S, Huey BD, Padture NP. Direct Observation of Ferroelectric Domains in Solution-Processed CH₃NH₃PbI₃ Perovskite Thin Films. *The Journal of Physical Chemistry Letters* 2014, **5**(19): 3335-3339.
7. Amano H, Asahi T, Akasaki I. Stimulated Emission Near Ultraviolet at Room Temperature from a GaN Film Grown on Sapphire by MOVPE Using an AlN Buffer Layer. *Japanese Journal of Applied Physics* 1990, **29**(Part 2, No. 2): L205-L206.
8. Ge J-F, Liu Z-L, Liu C, Gao C-L, Qian D, Xue Q-K, *et al.* Superconductivity above 100 K in single-layer FeSe films on doped SrTiO₃. *Nature materials* 2014, **14**: 285.
9. Wang Y, Sun X, Chen Z, Sun Y-Y, Zhang S, Lu T-M, *et al.* High-Temperature Ionic Epitaxy of Halide Perovskite Thin Film and the Hidden Carrier Dynamics. *Advanced Materials* 2017, **29**(35): 1702643.
10. Park N-G, Grätzel M, Miyasaka T, Zhu K, Emery K. Towards stable and commercially available perovskite solar cells. *Nature Energy* 2016, **1**: 16152.
11. Son D-Y, Lee J-W, Choi YJ, Jang I-H, Lee S, Yoo PJ, *et al.* Self-formed grain boundary healing layer for highly efficient CH₃ NH₃ PbI₃ perovskite solar cells. *Nature Energy* 2016, **1**: 16081.
12. Long R, Liu J, Prezhdo OV. Unravelling the Effects of Grain Boundary and Chemical Doping on Electron–Hole Recombination in CH₃NH₃PbI₃ Perovskite by Time-Domain Atomistic Simulation. *Journal of the American Chemical Society* 2016, **138**(11): 3884-3890.
13. Shao Y, Fang Y, Li T, Wang Q, Dong Q, Deng Y, *et al.* Grain boundary dominated ion migration in polycrystalline organic–inorganic halide perovskite films. *Energy & Environmental Science* 2016, **9**(5): 1752-1759.
14. Nie W, Tsai H, Asadpour R, Blancon J-C, Neukirch AJ, Gupta G, *et al.* High-efficiency solution-processed perovskite solar cells with millimeter-scale grains. *Science* 2015, **347**(6221): 522-525.
15. de Quilettes DW, Vorpahl SM, Stranks SD, Nagaoka H, Eperon GE, Ziffer ME, *et al.* Impact of microstructure on local carrier lifetime in perovskite solar cells. *Science* 2015, **348**(6235): 683-686.
16. Fang Y, Dong Q, Shao Y, Yuan Y, Huang J. Highly narrowband perovskite single-crystal photodetectors enabled by surface-charge recombination. 2015, **9**: 679.
17. Yang Y, Yan Y, Yang M, Choi S, Zhu K, Luther JM, *et al.* Low surface recombination velocity in solution-grown CH₃NH₃PbBr₃ perovskite single crystal. *Nature Communications* 2015, **6**: 7961.
18. Yang Y, Yang M, Moore David T, Yan Y, Miller Elisa M, Zhu K, *et al.* Top and bottom surfaces limit carrier lifetime in lead iodide perovskite films. *Nature Energy* 2017, **2**: 16207.
19. Leijtens T, Eperon GE, Barker AJ, Grancini G, Zhang W, Ball JM, *et al.* Carrier trapping and recombination: the role of defect physics in enhancing the open circuit voltage of metal halide perovskite solar cells. *Energy & Environmental Science* 2016, **9**(11): 3472-3481.

20. Yang WS, Park B-W, Jung EH, Jeon NJ, Kim YC, Lee DU, *et al.* Iodide management in formamidinium-lead-halide-based perovskite layers for efficient solar cells. *Science* 2017, **356**(6345): 1376-1379.
21. Ball JM, Petrozza A. Defects in perovskite-halides and their effects in solar cells. *Nature Energy* 2016, **1**: 16149.
22. Azpiroz JM, Mosconi E, Bisquert J, De Angelis F. Defect migration in methylammonium lead iodide and its role in perovskite solar cell operation. *Energy & Environmental Science* 2015, **8**(7): 2118-2127.
23. Rehman W, Milot RL, Eperon GE, Wehrenfennig C, Boland JL, Snaith HJ, *et al.* Charge-Carrier Dynamics and Mobilities in Formamidinium Lead Mixed-Halide Perovskites. *Advanced Materials* 2015, **27**(48): 7938-7944.
24. He H, Yu Q, Li H, Li J, Si J, Jin Y, *et al.* Exciton localization in solution-processed organolead trihalide perovskites. 2016, **7**: 10896.
25. Juarez-Perez EJ, Sanchez RS, Badia L, Garcia-Belmonte G, Kang YS, Mora-Sero I, *et al.* Photoinduced Giant Dielectric Constant in Lead Halide Perovskite Solar Cells. *The Journal of Physical Chemistry Letters* 2014, **5**(13): 2390-2394.
26. Zhu H, Miyata K, Fu Y, Wang J, Joshi PP, Niesner D, *et al.* Screening in crystalline liquids protects energetic carriers in hybrid perovskites. *Science* 2016, **353**(6306): 1409-1413.
27. Herz LM. Charge-Carrier Dynamics in Organic-Inorganic Metal Halide Perovskites. *Annual Review of Physical Chemistry* 2016, **67**(1): 65-89.
28. Shi D, Adinolfi V, Comin R, Yuan M, Alarousu E, Buin A, *et al.* Low trap-state density and long carrier diffusion in organolead trihalide perovskite single crystals. *Science* 2015, **347**(6221): 519-522.
29. Dong Q, Fang Y, Shao Y, Mulligan P, Qiu J, Cao L, *et al.* Electron-hole diffusion lengths >175 μm in solution grown $\text{CH}_3\text{NH}_3\text{PbI}_3$ single crystals. *Science* 2015.
30. Distler GI, Zvyagin BB. A Dislocation-free Mechanism of Growth of Real Crystals. *Nature* 1966, **212**(5064): 807-808.
31. Matthews JW, Blakeslee AE. Defects in epitaxial multilayers: I. Misfit dislocations. *Journal of Crystal Growth* 1974, **27**: 118-125.
32. Chen H, Guo LW, Cui Q, Hu Q, Huang Q, Zhou JM. Low - temperature buffer layer for growth of a low - dislocation - density SiGe layer on Si by molecular - beam epitaxy. *Journal of Applied Physics* 1996, **79**(2): 1167-1169.
33. Mendelson S. Dislocation Etch Pit Formation in Sodium Chloride. *Journal of Applied Physics* 1961, **32**(8): 1579-1583.
34. Chen J, Morrow DJ, Fu Y, Zheng W, Zhao Y, Dang L, *et al.* Single-Crystal Thin Films of Cesium Lead Bromide Perovskite Epitaxially Grown on Metal Oxide Perovskite (SrTiO_3). *Journal of the American Chemical Society* 2017, **139**(38): 13525-13532.
35. Malard LM, Pimenta MA, Dresselhaus G, Dresselhaus MS. Raman spectroscopy in graphene. *Physics Reports* 2009, **473**(5): 51-87.
36. Liang X, Sperling BA, Calizo I, Cheng G, Hacker CA, Zhang Q, *et al.* Toward Clean and Crackless Transfer of Graphene. *ACS Nano* 2011, **5**(11): 9144-9153.
37. Lu Z, Sun X, Xiang Y, Washington MA, Wang G-C, Lu T-M. Revealing the Crystalline Integrity of Wafer-Scale Graphene on SiO_2/Si : An Azimuthal RHEED Approach. *ACS Applied Materials & Interfaces* 2017, **9**(27): 23081-23091.
38. Sun X, Lu Z, Xiang Y, Wang Y, Shi J, Wang G-C, *et al.* van der Waals Epitaxy of Antimony Islands, Sheets, and Thin Films on Single-Crystalline Graphene. *ACS Nano* 2018, **12**(6): 6100-6108.
39. Yaffe O, Guo Y, Tan LZ, Egger DA, Hull T, Stoumpos CC, *et al.* Local Polar Fluctuations in Lead Halide Perovskite Crystals. *Physical Review Letters* 2017, **118**(13): 136001.

40. Zhou Y, Sternlicht H, Padture NP. Transmission Electron Microscopy of Halide Perovskite Materials and Devices. *Joule* 2019, **3**(3): 641-661.
41. Pernot C, Hirano A, Iwaya M, Detchprohm T, Amano H, Akasaki I. Low-Intensity Ultraviolet Photodetectors Based on AlGaIn. *Japanese Journal of Applied Physics* 1999, **38**(Part 2, No. 5A): L487-L489.

Reviewers' comments:

Reviewer #1 (Remarks to the Author):

In respect to my previous report the authors have changed and improved the manuscript to my satisfaction. I find the manuscript interesting and the ideas are sufficiently supported by experimental results. I recommend publication of the manuscript.

Reviewer #2 (Remarks to the Author):

Some discussion is still not crystal clear but distinguishes of the effect of many types of defects is challenging. I believe that now the manuscript is better to be judged by the readers as the authors provided a lot of data and discussion worth discussing. Therefore I recommend the manuscript for publication in Nature Communications.

Reviewer #3 (Remarks to the Author):

Thanks authors to try a great effort to address issues raised before. The manuscript is much improved. I would suggest to further clarify the following points before publication.

1. Please add a color scale bar for raman mapping data in Fig. R16, Fig. R18
2. Authors need to clarify the reason of having thick step height in Fig. R9
3. According to data that authors newly added regarding graphene quality (Fig. R19), graphene seems to have a lot of wrinkles where potentially have a lower activation energy for nucleation. It could have another role in kinetics. Authors need to include the clear explanation and supporting data for this effect
5. Authors need clarify why the photocurrent decay in both film (remote and ionic) shows similar value although film quality must be different.
6. In figure 4, authors included comparison between remote epitaxy and van der Waal epitaxy. It would be meaningful to include some experimental data to support this claim. It might give better crystallinity or film quality in remote epitaxy case if the potential field guides atomic registry compared to van der Waals epitaxy. Please consider to include comparison of crystallinity between those two cases.

Point-by-point response to Reviewers' comments (review comments are in black fonts and our replies are in blue fonts):

Reviewer #1 (Remarks to the Author):

In respect to my previous report the authors have changed and improved the manuscript to my satisfaction. I find the manuscript interesting and the ideas are sufficiently supported by experimental results. I recommend publication of the manuscript.

Our response:

We thank reviewer #1's recommendation for publication.

Reviewer #2 (Remarks to the Author):

Some discussion is still not crystal clear but distinguishes of the effect of many types of defects is challenging. I believe that now the manuscript is better to be judged by the readers as the authors provided a lot of data and discussion worth discussing. Therefore I recommend the manuscript for publication in Nature Communications.

Our response:

We thank reviewer #2's recommendation for publication.

Reviewer #3 (Remarks to the Author):

Thanks authors to try a great effort to address issues raised before. The manuscript is much improved. I would suggest to further clarify the following points before publication.

1. Please add a color scale bar for raman mapping data in Fig. R16, Fig. R18
2. Authors need to clarify the reason of having thick step height in Fig. R9
3. According to data that authors newly added regarding graphene quality (Fig. R19), graphene seems to have a lot of wrinkles where potentially have a lower activation energy for nucleation. It could have another role in kinetics. Authors need to include the clear explanation and supporting data for this effect
5. Authors need clarify why the photocurrent decay in both film (remote and ionic) shows similar value although film quality must be different.
6. In figure 4, authors included comparison between remote epitaxy and van der Waal epitaxy. It would be meaningful to include some experimental data to support this claim. It might give better crystallinity or film quality in remote epitaxy case if the potential field guides atomic registry compared to van der Waals epitaxy. Please consider to include comparison of crystallinity between those two cases.

Our response:

We thank the reviewer #3 for carefully reading the response letter and the supportive comments. Here we clarify the reviewer's comments point by point.

1. Please add a color scale bar for raman mapping data in Fig. R16, Fig. R18

Our response:

Following the advice, we have added all the color scale bars for Raman mapping data in Fig. S1 and S7 in the revised SI.

Associated changes to the main text and supplementary information (SI): Figures: Scale bars in Fig. S1 and S7 in the revised SI have been added.

2. Authors need to clarify the reason of having thick step height in Fig. R9

Our response:

The AFM Fig. S6a and b in SI (Fig. R9a and b in the previous response letter) show the morphology of CsPbBr₃ flakes grown by ionic epitaxy (CsPbBr₃/NaCl) and remote epitaxy (CsPbBr₃/Gr/NaCl), respectively. The “thick step heights” in them are the thickness of the corresponding **flakes** rather than the steps heights from a single domain. The AFM Fig. S6c and d in SI (Fig. R9c and d in the previous response letter) show the morphology of CsPbBr₃ thin film grown by ionic epitaxy (CsPbBr₃/NaCl) and remote epitaxy (CsPbBr₃/Gr/NaCl), respectively. The few nanometers step heights in them may be due to different domains merging together. Such (a few nanometers) RMS roughness is acceptable for most halide perovskite thin film cells (their thickness is often several hundred nanometers). In the revised manuscript, we have clarified this confusion.

3. According to data that authors newly added regarding graphene quality (Fig. R19), graphene seems to have a lot of wrinkles where potentially have a lower activation energy for nucleation. It could have another role in kinetics. Authors need to include the clear explanation and supporting data for this effect

Our response:

The optical microscopy image and Raman mapping images of a transferred graphene in Fig. R19 in the former response letter are picked up from an early literature¹ for comparison. Using a similar transfer process as that reported in the literature¹, our transferred graphene on NaCl and CaF₂ substrates show similar morphology in Fig. S1a and b in SI (Fig. R16a and b in the previous response letter) and there are wrinkles as well. We agree to the reviewer’s comment that the wrinkles (defected regions) of graphene could have lower activation energy for nucleation and might affect growth kinetics if we only consider graphene itself. However, at the wrinkles of graphene, the substrate-film coupling strength could be weaker due to the larger film-substrate distance compared to non-defective region since electrostatic

Figure R1 Optical microscopy image of remote epitaxial CsPbBr₃ flakes. The wrinkles of graphene (light gray) are indicated in red ellipse.

interaction (both van der Waals and ionic) decays as distance increases. In our case, the wrinkles of graphene can still be seen among the remote epitaxial CsPbBr₃ flakes after growth, as indicated in red ellipses in the optical microscopy image of Fig. R1 (Fig. S9c in SI). Our experimental observation indicates that the activation energy for nucleation at graphene/substrate surface seems to be even lower than that at wrinkles. Hence, in the remote epitaxy in our case, the growth kinetics might be dominated by the polar substrates while the wrinkles of graphene play a minor role. To clarify this, we have made corresponding edits in the revised SI.

5. Authors need clarify why the photocurrent decay in both film (remote and ionic) shows similar value although film quality must be different.

Our response:

As shown in Fig. S24d in former manuscript (now Fig. S25d in the revised version), they actually show different decay trends: an additional decay tail with a long decay time of around 8.08 s can be observed in the ionic epitaxial thin film-based device. The decay tail is probably related to the crystallinity/dislocation density in semiconductors, as reported in AlGaN based photodetectors². To clarify this, we have made corresponding edits in the manuscript.

6. In figure 4, authors included comparison between remote epitaxy and van der Waal epitaxy. It would be meaningful to include some experimental data to support this claim. It might give better crystallinity or film quality in remote epitaxy case if the potential field guides atomic registry compared to van der Waals epitaxy. Please consider to include comparison of crystallinity between those two cases.

Our response:

We completely agree to reviewer's points here. We appreciate the reviewer's suggestion that it would be meaningful to include some experimental data comparing the crystallinity between remote epitaxy and van der Waal (vdW) epitaxy. The Molecular-dynamics (MD) simulations in Fig. 4a-i in main text, Fig. S15-18 in SI and Video 4-9 compare the growth mechanism among ionic, remote and vdW epitaxy, revealing high crystallinity without dislocations in both remote and vdW epitaxy. We agree the reviewer's comment that remote epitaxy might yield better film quality due to the guide of potential field from the substrate compared to vdW epitaxy. For vdW epitaxy, indeed we have grown halide perovskite on a native oxide Si(100) substrate (non-polar) with transferred graphene on its surface (Gr/Si(100)). The growth result is shown in Fig. R2a (Fig. S22a in the revised SI). As expected, lower nucleation and poorer epitaxy have been found in pure vdW epitaxy of halide perovskite on graphene than remote epitaxy. This highlights the importance of extra potential guide of substrate in remote epitaxy. To improve substrate interaction, mica is chosen to substitute the Gr/Si(100) substrate. As shown in Fig. R2b and c (Fig. S22b and c in the revised SI), better crystal quality of halide perovskite (compared to Gr/Si) has been achieved in our growth. However, due to the in-plane symmetry mismatch between halide perovskite and mica, the epitaxial halide perovskite flake shows in-plane rotation as shown in Fig. R2d. Without the guide of potential field with matched symmetry from the polar substrate, the in-plane film quality in quasi-vdW epitaxy is lower than that in remote epitaxy. However, the out-of-plane crystallinity of halide perovskite is better for the van der Waals epitaxy case compared to that for remote epitaxy, as shown in XRD ω -2 θ scanning and rocking curve in Fig. R2e and f (Fig. S22e and f in the revised SI), respectively.

To clarify these, associated changes to the main text and supplementary information (SI) have been made. Figures: Fig. S22 on the vdW epitaxy of halide perovskite has been added the revised SI. The ordinal numbers of figures after Fig. S22 (Fig. S23-27) has been revised in both revised SI and main text.

Figure R2 vdW epitaxy of halide perovskite. Optical microscopy images of halide perovskite flakes grown on Gr/Si(100) (a) and mica (b-d), and XRD θ - 2θ scanning of CsPbBr₃/Mica (e) and rocking curve of CsPbBr₃ 004 (f).

Response References

- [1] X. Li, Y. Zhu, W. Cai, M. Borysiak, B. Han, D. Chen, R.D. Piner, L. Colombo and R.S. Ruoff, "Transfer of Large-Area Graphene Films for High-Performance Transparent Conductive Electrodes", *Nano Letters*, **9**, 4359-4363 (2009)
- [2] C. Pernot, A. Hirano, M. Iwaya, T. Detchprohm, H. Amano and I. Akasaki, "Low-Intensity Ultraviolet Photodetectors Based on AlGaN", *Japanese Journal of Applied Physics*, **38**, L487-L489 (1999)

Reviewers' comments:

Reviewer #3 (Remarks to the Author):

We are satisfied with modified version of the manuscript. After addressing minor comments below, I would recommend the publication of this work.

1. I still believe that authors would better change the title. Although authors added some data regarding localized place, kinetic must be really complicated and other defects still contribute to the phenomenon. Until we see them, it is really hard to know how they play a role in the film. Accordingly, instead of mentioning dislocation in the title again, it would be much better to use something else like 'carrier life time enhancement via remote epitaxy'. Also, in the manuscript, some statement could be modified or new statements should be added to explain this point of view regarding other defects.
2. Authors need a color scale bar for raman mapping data in Fig. R16, Fig. R18
3. Authors need to clarify the reason having thick step height in Fig. R9
4. According to data that authors newly add data regarding graphene quality (Fig. R19), graphene seems to have a lot of wrinkle where potentially have a lower activation energy for nucleation. It could have another role in kinetic. I believe that if authors include the explanation and supporting data for this effect, it would be useful for readers to understand phenomenon
5. Authors clarify why the photocurrent decay in both film (remote and ionic) shows similar value although film quality must be different.
6. In figure 4, authors newly include comparison between remote epitaxy and van der Waal epitaxy. It would be meaningful to include some experimental data to support this claim. It might give better crystallinity or film quality in remote epitaxy case if the potential field guides atomic registry compared to van der Waals epitaxy. If authors include crystallinity comparison experimentally, it could be really interesting.
7. What I asked was not scanning transmission electron microscopy but scanning tunneling electron microscopy to observe the potential field. STEM might scan the potential penetration. It could be interesting to see the potential field comparison.
8. Why does graphene show only little color contrast only in STEM?
9. In p39, authors claimed that "in our case, grain size, if there is, and interfaces are tried to be kept constant during characterization". I know the authors' point but I believe that it is hard to claim like that until we see that experimentally.

Response to Reviewers' comments (review comments are in black fonts and our replies are in blue fonts):

Reviewer #3 (Remarks to the Author):

Manuscript is very well shaped. I thank authors to upgrade the manuscript so much. Now ready to be published. One recommendation for authors is to modify the title as suggested before like also suggested by another reviewer.

Our response:

We thank the reviewer #3 for the supportive comments. We have modified the title as suggested, which is “Carrier Lifetime Enhancement in Halide Perovskite via Remote Epitaxy”.